# Controlling Federated Learning for Covertness

**Adit Jain**                                                                 *aj457@cornell.edu*
*Department of Electrical and Computer Engineering, Cornell University*

**Vikram Krishnamurthy**                                                      *vikramk@cornell.edu*
*Department of Electrical and Computer Engineering, Cornell University*

**Reviewed on OpenReview:** *https://openreview.net/forum?id=g01OVahtN9*

## Abstract

A learner aims to minimize a function $f$ by repeatedly querying a distributed oracle that provides noisy gradient evaluations. At the same time, the learner seeks to hide $\arg\min f$ from a malicious eavesdropper that observes the learner's queries. This paper considers the problem of *covert* or *learner-private* optimization, where the learner has to dynamically choose between learning and obfuscation by exploiting the stochasticity. The problem of controlling the stochastic gradient algorithm for covert optimization is modeled as a Markov decision process, and we show that the dynamic programming operator has a supermodular structure implying that the optimal policy has a monotone threshold structure. A computationally efficient policy gradient algorithm is proposed to search for the optimal querying policy without knowledge of the transition probabilities. As a practical application, our methods are demonstrated on a hate speech classification task in a federated setting where an eavesdropper can use the optimal weights to generate toxic content, which is more easily misclassified. Numerical results show that when the learner uses the optimal policy, an eavesdropper can only achieve a validation accuracy of $52\%$ with no information and $69\%$ when it has a public dataset with $10\%$ positive samples compared to $83\%$ when the learner employs a greedy policy.

## 1 Introduction

### 1.1 Main Results

A learner aims to minimize a function $f$ by querying an oracle repeatedly. At times $k = 0, 1, \ldots$, the learner sends a query $q_k$ to the oracle, and the oracle responds with a noisy gradient evaluation $r_k$. Ideally, the learner would use this noisy gradient in a stochastic gradient algorithm to update its estimate of the minimizer, $\hat{x}_k$ as: $\hat{x}_{k+1} = \hat{x}_k - \mu_k\, r_k$, where $\mu_k$ is the step size and pose the next query as $q_{k+1} = \hat{x}_{k+1}$. However, the learner seeks to hide the $\arg\min f$ from an eavesdropper. The eavesdropper observes the sequence of queries $(q_k)$ but does not observe the responses from the oracle. The eavesdropper is passive and does not directly affect the queries or the responses. How can the learner perform stochastic gradient descent to learn $\arg\min f$ but hide it from the eavesdropper? This problem arises in federated learning (FL), where the central learner (e.g. an application) optimizes the loss function of a neural network by communicating with a distributed set of devices. The learner communicates the weights to the devices and receives noisy gradients of the loss function evaluated on local data which the learner uses to update the weights.

Our proposed approach is to control the stochastic gradient descent (SGD) using another stochastic gradient algorithm, namely a structured policy gradient algorithm (SPGA) that solves a resource allocation Markov decision process. The following two-step cross-coupled stochastic gradient algorithm summarizes our approach:

$$
\begin{aligned}
\text{Stochastic Gradient Descent: } & \hat{x}_{k+1} = \hat{x}_k - \mu_k G(r_k, y_k, q_k), \\
\text{Query using Policy from SPGA: } & q_{k+1} \sim P(\nu(y_{k+1}), \hat{x}_{k+1}).
\end{aligned}
\tag{1}
$$

Here $k$ is the time index, $\mu_k$ is the step size, $q_k$ is the query and $y_k$ is the system state. The function $G$ is designed by the learner to update the estimate based on the noisy response $r_k$ (for example, $G$ can be 0 when obfuscating and the

noisy gradient $r_k$ otherwise). $P$ is the probability distribution of the query based on the transition probability kernel of the stationary policy, which decides whether to learn or obfuscate. The first equation in (1) updates the learner's $\arg\min$ estimate, $\hat{x}_k$, and the second equation computes the next query $q_{k+1}$ using a policy $\nu$.

**Contributions:**

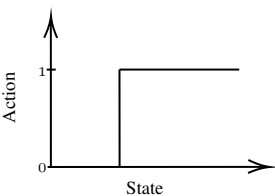

(a) Threshold structure of the optimal policy with two actions.

- This paper proposes a framework for the learner to dynamically query the oracle for robust learning and covert optimization. We formulate a Markov decision process (MDP) to solve the decision problem of the learner by exploiting the inherent stochasticity of the oracle. Structural results which show that the optimal policy has a threshold structure (Fig. 1(a)) are proven in Theorem 2 and Theorem 3. The structural results enable the use of efficient policy search methods to search for the optimal policy. This framework can be extended to meta-learning problems like controlling learning for energy efficiency and quality control.

- A policy gradient algorithm is proposed which estimates the optimal stationary policy for the MDP. The optimal stationary policy controls the primary stochastic gradient descent of the learner as described by (1) and shown in Fig. 1(b). The policy gradient algorithm has a linear time complexity due to the threshold nature of the optimal policy and does not need knowledge of the transition probabilities. The policy gradient algorithm runs on a slower time scale and can adapt to changes in the system.

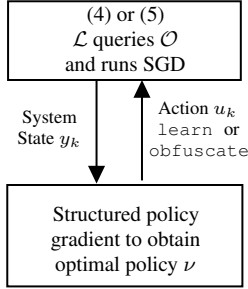

(b) Coupled interaction of the two gradient algorithms.

Figure 1: To achieve covert optimization, we propose controlling the SGD by a policy gradient algorithm that exploits the policy structure.

- The proposed methods are demonstrated on a novel application, covert federated learning (FL) on a text classification task using large language model embeddings. Our key numerical results are summarized in Table 2. It is shown that compared to a greedy policy when the learner is using the optimal policy, the the eavesdropper's estimate of the optimal weights do not generalize well, even if the eavesdropper uses a public dataset to validate the queries [1].

- This paper considers two pragmatic aspects of empirical distributed learning environments: a stochastic oracle and an optimization queue. A stochastic oracle with different noise levels can model a non-i.i.d. client distribution in FL, and an optimization queue can model repeated optimization tasks like machine unlearning and distribution shifts. Theorem 1 characterizes the number of successful updates required for convergence, and Lemma 1 analyzes the stability of the queue.

## 1.2 Motivation and Related Works

The main application of covert stochastic optimization is in distributed optimization where the central learner queries a distributed oracle and receives gradient evaluations using which it optimizes $f$. Such a distributed oracle is part of federated learning where deep learning models are optimized on mobile devices with distributed datasets (McMahan et al., 2017) and pricing optimization where customers are surveyed (Delahaye et al., 2017). In distributed optimization, the eavesdropper can pretend as a local worker of the distributed oracle and obtain the queries posed (but not the aggregated responses). The eavesdropper can infer the local minima by observing the queries. These estimates can be used for malicious intent or competitive advantage since obtaining reliable, balanced, labeled datasets is expensive. As an example, in hate speech classification in a federated setting (Meskys et al., 2019; Narayan et al., 2022), a hate

---

[1]A greedy policy poses learning queries until all the required successful updates are done to the model and then starts posing obfuscating queries.

| Symbol | Description |
|--------|-------------|
| $k$ | Time index for the stochastic gradient descent (SGD) |
| $n$ | Index for stochastic policy gradient algorithm (SPGA) |
| $\hat{x}$ | Learner's estimate of the minima |
| $\hat{z}$ | Eavesdropper's estimate of the minima |
| $y$ | State variable for the oracle state, learner state and arrival state |
| $q$ | Query posed to the oracle by the learner |
| $u$ | Action of the learner (Learning or Obfuscation) |
| $r$ | Noisy gradient evaluation by oracle of function at query point |
| $\sigma_k$ | Bound on the variance of the noise added at time $k$ |
| $\sigma$ | Tolerance parameter of learner for bound on noise variance |
| $\phi$ | True threshold parameter of the optimal policy |
| $\theta$ | Threshold parameter of the SPGA tracking the true threshold parameter |
| $\mu, \kappa$ | Step size of the SGD and SPGA respectively |

Table 1: Summary of mathematical notations used in this paper.

speech peddler can pretend as a client and observe the trajectory of the learner to obtain the optimal weights minimizing the loss function. Using these optimal weights for discrimination, the eavesdropper can train a generator network to generate hate speech which will not be detected (Hartvigsen et al., 2022; Wang et al., 2018).

**Learner-private or covert optimization:** The problem of protecting the optimization process from eavesdropping adversaries is examined in recent literature (Xu et al., 2021a; Tsitsiklis et al., 2021; Xu, 2018; Xu et al., 2021b). Tsitsiklis et al. (2021); Xu et al. (2021b;a) to obtain $(L, \delta)$-type privacy bounds on the number of queries required to achieve a given level of learner-privacy and optimize with and without noise. The current state of art (Xu et al., 2021a) in covert optimization dealt with convex functions in a noisy setting. Although there is extensive work in deriving the theoretical proportion of queries needed for covert optimization, practically implementing covert optimization has not been dealt with. In contrast, our work considers a non-convex noisy setting with the problem of posing learning queries given a desired level of privacy. We want to schedule $M$ model updates in $N$ queries and obfuscate otherwise, ideally learning when the noise of the oracle is expected to be less. Theoretical bounds are difficult to derive for a non-convex case, but we empirically show our formulation achieves covertness. The approach of this work aligns with hiding optimal policies in reinforcement learning when actions are visible (Liu et al., 2021; Pan et al., 2019; Dethise et al., 2019), information theory (Bloch, 2016), and preserving privacy while traversing graphs (Erturk & Xu, 2020) [2].

**Motivation for stochastic oracle and optimization queue:** An important application of covert optimization is in federated learning, which is deployed in various machine learning tasks to improve the privacy of dataset owners and communication efficiency (Heusel et al., 2017; Chen et al., 2022a). FL involves non-i.i.d. data distribution across clients and time-varying client participation and data contribution which motivate a dynamic stochastic oracle (Karimireddy et al., 2020; Jhunjhunwala et al., 2022; Doan et al., 2020; Chen et al., 2022b). Similar to recent work in a setup with

---

[2]Distinction from differential privacy: Differential privacy Lee et al. (2021); Dimitrov et al. (2022); Asi et al. (2021) is concerned with the problem of preserving the privacy of the local client's data by mechanisms like adding noise and clipping gradients, whereas learner-private or covert optimization is used when the central learner is trying to prevent an eavesdropper from freeriding on the optimization process (Tsitsiklis et al., 2021).

| | Scenario 1 | | Scenario 2 | |
|---|---|---|---|---|
| | Eavesdropper has no toxic samples | | Eavesdropper has toxic samples | |
| Type of Policy | Eavesdropper accuracy | Learner accuracy | Eavesdropper accuracy | Learner accuracy |
| Greedy | 0.83 | 0.84 | 0.83 | 0.82 |
| Optimal | 0.52 | 0.81 | 0.69 | 0.81 |

Table 2: The optimal policy to our MDP formulation achieves covertness: The eavesdropper's accuracy is reduced significantly in comparison to a greedy policy (-31%) even when the eavesdropper has samples to validate the queries (-14%). The learner's accuracy remains comparable in both the scenarios with a maximum difference of 3%.

Markovian noise in FL (Sun et al., 2018; Rodio et al., 2023), we model the bounds on the variance of the oracle noise to vary as a discrete Markov chain, and exploit the Markovian nature of the oracle's stochasticity to query dynamically. An optimization queue for training tasks is motivated by active learning where the learner waits for training data to be annotated and performs optimization once there is enough annotated data (Bachman et al., 2017; Wu et al., 2020). Optimization tasks can also be due to purging client's data, training in new contexts, distributional shifts, or by design of the learning algorithm (Tripp et al., 2020; Cai et al., 2021; Mallick et al., 2022; Ginart et al., 2019; Sekhari et al., 2021).

### 1.3   Organization

In Section 2, the problem of minimizing a function while hiding the $\arg\min$ is modeled as a controlled SGD. In Section 3, a finite horizon MDP is first formulated where the learner has to perform $M$ successful gradient updates of the SGD in $N$ total queries to a stochastic oracle. The formulation is extended to an infinite horizon constrained MDP (CMDP) for cases when the learner performs optimization multiple times, and a policy gradient algorithm is proposed to search for the optimal policy. Numerical results are demonstrated on a classification task in Section 4. A discussion of the stochastic oracle, dataset description, additional experimental results and proofs, are presented in the Appendix. Notations are summarized in Table 1.

**Limitations:**  The assumption of unbiased responses and the independence of the eavesdropper and oracle might not hold, especially if the eavesdropper can deploy multiple devices. Due to the lack of data on device participation in FL, it is difficult to verify the assumption of a Markovian oracle. The assumption on an equal eavesdropper prior for both the obfuscating and learning SGD has not been theoretically verified. The assumption that the learner knows about the eavesdropper's dataset distribution might only hold if it is a public dataset or if there are obvious costly classes.

## 2   Controlled Stochastic Gradient Descent for Covert Optimization

This section discusses the oracle, the learner, and the eavesdropper and states the assumptions which are essential in modeling the problem as a Markov decision process in the next section. The following is assumed about the oracle $\mathcal{O}$:

(A1): (**Bounded from below and Lipschitz continous**) The oracle is a function $f : \mathbb{R}^d \to \mathbb{R}$. $f$ is continous and is bounded from below, $f \geq f^*$. $f$ is continuously differentiable and its derivative is $\gamma$-Lipschitz continous, i.e. $\|\nabla f(z) - \nabla f(x)\| \leq \gamma \|z - x\| \, \forall x, z \in \mathbb{R}^d$, where $\nabla f$ indicates the gradient of $f$ and $\|\cdot\|$ is the $l^2$-norm.

At time $k$, for a query $q_k \in \mathbb{R}^d$, the oracle returns a noisy evaluation $r_k$ of $\nabla f$ at $q_k$ with added noise $\eta_k$,

$$r_k(q_k) = \nabla f(q_k) + \eta_k. \tag{2}$$

(A2): (**Assumption on Noise**) The noise $\eta_k$ is such that $r_k(q_k)$ is an unbiased estimator of $\nabla f(q_k)$, $\mathbb{E}[r_k(q_k)] = \nabla f(q_k)$ . The noise has a bounded variance $\mathbb{E}\left[\|\eta_k\|^2\right] \leq (\sigma_k)^2$. $\sigma_k^2$ is a constant that the oracle returns along the response, $r_k(q_k)$.

Assumptions (A1) and (A2) are regularity assumptions for analyzing query complexity results of the stochastic gradient descent, and are found in standard first-order gradient descent analysis (Ghadimi & Lan, 2013; Ajalloeian & Stich, 2021). There assumption on the noise terms is slightly weaker than independence (Ghadimi & Lan, 2013). Our analysis can be extended to algorithms with better convergence rates like momentum based Adam (Kingma & Ba, 2017). Besides simplicity in exposition, SGD is shown to have better generalizability  Zhou et al. (2020); Wilson et al. (2017). $f$ can be an empirical loss function for a training dataset $D$, $f(x; D) = \sum_{d_i \in D} \mathcal{G}(x; d_i)$, $D = \{d_i\}$ where $\mathcal{G}(\cdot; \cdot)$ is a loss function. In FL, the oracle is a collection of clients acting as a distributed dataset and compute.

The objective of the learner $\mathcal{L}$ is to query the oracle and obtain a $\epsilon$-close estimate $\hat{x}$ such that,

$$\mathbb{E}\left(\|\nabla f(\hat{x})\|^2\right) \leq \epsilon, \tag{3}$$

given an initial estimate, $\hat{x}_0 \in \mathbb{R}^d$. The expectation $\mathbb{E}[\cdot]$ here is with respect to the noise sequence and the external random measure (if any) used to compute the estimate. The learner has knowledge of the Lipschitz constant $\gamma$ from (A1). At time $k$ for query $q_k$ the learner receives $(r_k, \sigma_k^2)$ from the oracle (the response $r_k(q_k)$ is denoted as $r_k$). The

learner iteratively queries the oracle with a sequence of queries $q_1, q_2, \ldots$ and correspondingly updates its estimate $\hat{x}_1, \hat{x}_2, \ldots$ for estimating $\hat{x}$ such that it achieves its objective (3).

In classical SGD, the learner iteratively updates its estimate based on the gradient evaluations at the previous estimate. Now, since the queries are visible and the learner has to obfuscate the eavesdropper, the learner can either query using its true previous estimate or obfuscate the eavesdropper as described later. The learner updates its estimates $\hat{x}_1, \hat{x}_2, \ldots$ based on whether the posed query $q_k$ is for learning or not and the received noise statistic $\sigma_k$. A learning query is denoted by action $u_k = 1$ and an obfuscating query by action $u_k = 0$. The learner chooses a noise constant $\sigma$ [3] and performs a controlled SGD with step size $\mu_k$ for $m^{\text{th}}$ successful update) such that it updates its estimate only if $\sigma_k^2 \leq \sigma^2$ and if a learning query was posed, i.e., $u_k = 1$ ($\mathbb{1}$ denotes the indicator function),

$$\hat{x}_{k+1} = \hat{x}_k - \mu_k r_k \mathbb{1}\left(\sigma_k^2 \leq \sigma^2\right) \mathbb{1}\left(u_k = 1\right). \tag{4}$$

For formulating the MDP in the next section, we need the following definition and theorem, which characterizes the finite nature of the optimization objective of (3). The proof of the theorem follows by applying the gradient lemma and standard convergence results to the update step (Bottou, 2004; Ghadimi & Lan, 2013; Ajalloeian & Stich, 2021) and is given in the Appendix B.1.

**Definition 1.** *Successful Update*: *At time $k$ an iteration of (4) is a successful update if $\mathbb{1}\left(\sigma_k^2 \leq \sigma^2\right) \mathbb{1}\left(u_k = 1\right) = 1$.*

**Theorem 1.** *(**Required number of successful updates**) Learner $\mathcal{L}$ querying an oracle $\mathcal{O}$ which satisfies assumptions (A1-2) using a controlled stochastic gradient descent with updates of the form (4) with a constant step size $\mu_k = \mu = \min\left\{\frac{\epsilon}{2\gamma\sigma^2}, \frac{1}{\gamma}\right\}$, needs to perform $M$ successful updates (Def. 1) to get $\epsilon$-close to a critical point (3), where*

$$M = O\left(\frac{1}{\epsilon} + \frac{\sigma^2}{\epsilon^2}\right).$$

Let the $M$ successful updates happen at time indices $k_1, \ldots, k_M$, then the learner's estimate of the critical point, $\hat{x}$ is chosen as $\hat{x}_{k_i}$ with probability $\frac{\mu_{k_i}}{\sum_1^M \mu_{k_j}}$, which is a uniform distribution for a constant step size.

Theorem 1 helps characterize the order of the number of successful gradient updates that need to be done in the total communication slots available to achieve the learning objective of (3). If the optimization is not one-off, the learner maintains a queue of optimization tasks where each optimization task requires up to order $M$ successful updates.

The obfuscation strategy used by the learner builds upon the strategy suggested in existing literature (Xu et al., 2021a; Xu, 2018). The learner queries either using the correct estimates from (4) or queries elsewhere in the domain to obfuscate the eavesdropper. In order to compute the obfuscating query, we propose that the learner runs a parallel SGD, which also ensures that the eavesdropper does not gain information from the shape of two trajectories. The obfuscating queries are generated using a suitably constructed function, $H$, and running a parallel SGD with an estimate, $\hat{z}_k$,

$$\hat{z}_{k+1} = \hat{z}_k - \mu_k H(r_k, \sigma_k, u_k). \tag{5}$$

At time $k$ an eavesdropper $\mathcal{E}$ has access to the query sequence, $(q_1, \ldots, q_k)$ which the eavesdropper uses to obtain an estimate, $\hat{z}$ for the $\arg\min f$. We make the following assumption on the eavesdropper:

(**E1**) The eavesdropper is assumed to be passive, omnipresent, independent of the oracle, and unaware of the chosen $\epsilon$.

(**E2**) Given a sequence of $N$ queries $(q_1, q_2, \ldots, q_N)$ which the eavesdropper observes, we assume that the eavesdropper can partition query sequence into two disjoint SGD trajectory sequences $I$ and $J$ [4].

(**E3**) In the absence of any additional information, the eavesdropper uses a proportional sampling estimator similar to Xu (2018) and is defined below.

---

[3] The noise constant $\sigma$ helps characterize the number of queries required and decide if a received response will be used for learning or not, this is in principle similar to the controlling communication done in Chen et al. (2022a); Sun et al. (2022). The probability of $\sigma_k \leq \sigma$ depends on the oracle state (defined in the next section). Using such a construction our framework enables characterizing an oracle which has varying noise levels.

[4] The parameter space is high dimensional, and it is assumed that SGD trajectories do not intersect. The final weights used in production can be transmitted in a secure fashion (using a much costlier, less efficient communication (Xu et al., 2023; Kairouz et al., 2021)).

Assumption (E1) is generally not true, especially if the eavesdropper is part of the oracle (for example, in FL), but we take this approximation assuming since the number of clients is much greater than the single eavesdropper and hence the oracle is still approximately Markovian. Given an equal prior over disjoint intervals, an eavesdropper using a proportional sampling estimator calculates the posterior probability of the minima lying in an interval proportional to the number of queries observed in the interval. Since this work considers two disjoint SGD trajectories, the eavesdropper's posterior probability of the minima belonging to one of the SGD trajectories given equal priors is proportional to the number of queries in the trajectories,

**Definition 2.** *Proportional sampling estimator: If the eavesdropper (with assumptions E1-2) observes queries belonging to two SGD trajectory sequences I and J and has equal prior over both of them, then the eavesdropper's posterior belief of the learner's true estimate $\hat{x}$ belonging to I is given by, $P_I \triangleq \mathbb{P}(\hat{x} \in I | (q_1, q_2, \ldots, q_N)) = \frac{|I|}{|I|+|J|}$.*

Let $K^*$ be defined as $K^* = \arg\max_{K \in \{I, J\}} P_K$ and $B[-1]$ retrieve the last item of the sequence $B$. After $N$ observed queries, the eavesdropper's maximum a posteriori estimate, $\hat{z}$ of the learner's true estimate, $\hat{x}$, is given by,

$$\hat{z} = (q_k | q_k \in K^*, k = 1, \ldots, N)[-1]. \tag{6}$$

An eavesdropper using a proportional sampling estimator with an estimate of the form (6) can be obfuscated by ensuring a) that the eavesdropper has equal priors over the two trajectory and b) that the majority of the queries posed are obfuscating [5]. Rather than posing the learning queries randomly, we formulate an MDP in the next section, which exploits the stochasticity of the oracle for optimally scheduling the queries.

# 3 Controlling the Stochastic Gradient Descent using a Markov decision process

This section formulates a Markov decision process that the learner $\mathcal{L}$ solves to obtain a policy to dynamically choose between posing a learning or an obfuscating query. The MDP is formulated for a finite horizon and infinite horizon case, and we prove structural results characterizing the monotone threshold structure of the optimal policy.

## 3.1 Finite Horizon Markov Decision Process

In the finite horizon case where the learner wants to optimize once and needs $M$ successful updates (obtained from Theorem 1 or chosen suitably) to achieve (3) with $N(> M)$ queries. Such a formulation helps model covert optimization of the current literature for a one-off FL task or a series of training tasks carried out one after the other.

The state space $\mathcal{S}$ is an augmented state space of the oracle state space $\mathcal{S}_O$ and the learner state space $\mathcal{S}_L$. The oracle is modeled to have $W$ oracle states, $\mathcal{S}_O = \{1, 2, \ldots W\}$. The learner state space has $M$ states, $\mathcal{S}_L = \{1, 2, \ldots, M\}$ which denote the number of successful updates left to be evaluated by the learner. The augmented state space is $\mathcal{S} = \mathcal{S}_O \times \mathcal{S}_L$. The state space variables with $n$ queries remaining (out of $N$) is denoted by $y_n = (y_n^O, y_n^L)$. As described in the obfuscation strategy, the learner can query either to learn using (4) or to obfuscate using (5), the action space is $\mathcal{U} = \{0 = \texttt{obfuscate}, 1 = \texttt{learn}\}$. $u_n$ denotes the action when $n$ queries are remaining .

$\Upsilon : \mathcal{S}_O \times \mathcal{U} \to [0, 1]$ denotes the probability of a successful update of (4) and is dependent on $y_n^O$ and action $u_n$, $\Upsilon(y_n^O, u_n) = \mathbb{P}\left(\sigma_n^2 \leq \sigma^2 \mid y_n^O\right) \mathbb{1}\left(u_n = 1\right)$. Hence, the bounds on the noise variance in (A2) of the SGD are dynamic and modeled as a finite state Markov chain. The learner state reduces by one if there is a successful update of (4), hence the transition probability between two states of the state space $\mathcal{S}$ is given by,

$$\mathbb{P}(y_{n-1}|y_n, u_n) = \mathbb{P}(y_{n-1}^O|y_n^O)\left(\Upsilon(y_n^O, u_n)\mathbb{1}(y_{n-1}^L = y_n^L - 1) + (1 - \Upsilon(y_n^O, u_n))\mathbb{1}(y_{n-1}^L = y_n^L)\right).$$

With $n$ queries left, for action, $u_n$, the learner incurs a privacy cost, $c(u_n, y_n^O)$ which accounts for the increase in the useful information known to the eavesdropper. At the end of $N$ queries, the learner incurs a terminal learning cost $l(y_0^L)$ which penalizes the remaining number of successful updates $y_0^L$. The following assumptions are made about the MDP:

**(M1)** The rows of the probability transition matrix between two states of the oracle are assumed to be first-order stochastic dominance orderable (Eq. (1) of Ngo & Krishnamurthy (2009)), i.e., $\sum_{k \geq l} \mathbb{P}(y_{n+1}^O = k | y_n^O = j) \leq$

---

[5]This can be extended to obfuscating with multiple intervals by using auxiliary trajectories and obfuscating by choosing uniformly between those.

$\sum \mathbb{P}(y_{n+1}^O = k | y_n^O = i) \, \forall \, i > j, \, l = 1, \ldots, W$. This is a restatement of the assumption that an oracle in a better state is more likely to stay in a better state, which is found to be empirically true for most computer networks.

(**M2**) $c(u_n, y_n^O)$ is chosen to decrease in $u_n$ and also decrease in oracle cost $y_n^O$ to incentivize learning when the oracle is in a good state. The learner does not incur a privacy cost when it does not learn, i.e., $c(0, y_n^O) = 0$.

(**M3**) $l(y_0^L)$ is increasing in $y_0^L$, integer convex, $l(y_0^L + 2) - l(y_0^L + 1) > l(y_0^L + 1) - l(y_0^L)$ and $l(0) = 0$.

(**M4**) With $n$ queries remaining, the learner has information on $y_n$, and eavesdropper does not have any information.

For a FL setting, (M1) assumes that if the client participation is high, it is less likely to drop suddenly. (M2-M3) ensure that the optimal policy prioritizes learning in a better oracle state, i.e., when the variance is more likely to be bounded. Integer convexity from (M3) in prove the structural results and ensures that the learning is prioritized more when the learner queue state is larger. The asymmetry in information in (M4) can be explained by (E1) since the eavesdropper is assumed to be a part of a much larger oracle. The objective for the finite horizon MDP can be formulated as minimizing the state action cost function $Q_n(y, u)$,

$$V_n(y) = \min_{u \in \mathcal{U}} Q_n(y, u), \quad \forall \, y \in \mathcal{S}, \tag{7}$$

where, $Q_n(y, u) = c(u_n, y^O) + \sum_{y' \in \mathcal{S}} P(y'|y, u_n) V_{n-1}(y')$ and $V_n(y)$ is the value function with $n$ queries remaining and $V_0(y) = l(y_0^L)$. The optimal decision rule is given by, $u_n^*(y) = \arg\min_{u \in \mathcal{U}} Q_n(y, u)$. The optimal policy $\nu^*$ is the sequence of optimal decision rules, $\nu^*(y) = (u_N^*(y), u_{N-1}^*(y), \ldots, u_1^*(y))$.

## 3.2 Infinite Horizon Constrained Markov Decision Process

This subsection formulates the infinite horizon constrained MDP (CMDP), highlights the key differences from the finite horizon case, introduces the optimization queue, and proves its stability. The CMDP formulation minimizes the average privacy cost while satisfying a constraint on the average learning cost. An optimization queue helps model the learner performing optimization repeatedly, which is needed for purging specific data, distributional shifts, and active learning.

**Optimization Queue:** The learner maintains a queue with the number of successful updates it needs to make, $y_n^L$ at time $n$. In contrast to the finite horizon case, the learner receives new requests and appends the optimization tasks to its queue. At time $n$, $y_n^E$ new successful updates required are added to the queue. $y_n^E$ is an i.i.d. random variable with $\mathbb{P}(E_n = M) = \delta$, $\mathbb{P}(E_n = 0) = 1 - \delta$ and $\mathbb{E}(E_n) = \delta M$ [6]. In order to ensure the queue is stable, we pose conditions on the success function $\Upsilon$, the learning cost $l$, and the learning constraint $\Lambda$ in Lemma 1.

The state space for the new arrivals to the queue is $\mathcal{S}_E = \{0, M\}$. Also, the learner state is denumerable, $\mathcal{S}_L = \{0, 1, \ldots\}$. The oracle state space, $\mathcal{S}_O$ is the same as before. The state space is now, $\mathcal{S} = \mathcal{S}_O \times \mathcal{S}_L \times \mathcal{S}_E$. The state variables at time $n$ ($n$ denotes the time index for deciding this section) are given by $y_n = (y_n^O, y_n^L, y_n^E)$.

The transition probability now has to incorporate the queue arrival, with the same $\Upsilon$ as before and can be written as,

$$\begin{aligned}
\mathbb{P}(y_{n+1}|y_n, u_n) = \mathbb{P}(y_{n+1}^O|y_n^O)\mathbb{P}(y_{n+1}^E) \, &\big(\Upsilon(y_n^O, u_n)\mathbb{1}(y_{n+1}^L = y_n^L + y_n^E - 1) \\
&+ (1 - \Upsilon(y_n^O, u_n))\mathbb{1}(y_{n+1}^L = y_n^L + y_n^E)\big).
\end{aligned} \tag{8}$$

The learning cost in the infinite horizon case is redefined as $l(u_n, y_n^O)$ and is decreasing in $u_n$ and increasing in $y_n^O$ which contrasts with $c$. The learning cost does not depend on $y_n^L$ except when $y_n^L = 0$, $l(u_n, y_n^O) = 0 \, \forall u_n \in \mathcal{U}, y_n^O \in \mathcal{S}_O$. The privacy cost $c(u_n, y_n^O)$, assumptions (M1-2), and the action space $\mathcal{U}$ are the same as the finite horizon case.

In the infinite horizon case, a stationary policy is a map from the state space to the action space, $\nu : \mathcal{S} \to \mathcal{U}$. Hence the policy generates actions, $\nu = (u_1, u_2, \ldots)$. Let $\mathcal{T}$ denote the space of all stationary policies. The average privacy cost and the average learning cost, respectively, are,

$$C_{y_0}(\nu) = \limsup_{N \to \infty} \frac{1}{N} \, \mathbb{E}_\nu \left[ \sum_{n=1}^N c(u_n, y_n^O) \mid y_0 \right], \quad L_{y_0}(\nu) = \limsup_{N \to \infty} \frac{1}{N} \, \mathbb{E}_\nu \left[ \sum_{n=1}^N l(u_n, y_n^O) \mid y_0 \right].$$

---

[6]The construction of $y^E$ and the i.i.d. condition is for convenience and we only require that $y^E$ is independent of the learner and the oracle state.

The constrained MDP can then be formulated as,

$$\inf_{\nu \in \mathcal{T}} C_{y_0}(\nu) \ \text{ s.t. } \ L_{y_0}(\nu) \le \Lambda \ \forall y_0 \in \mathcal{S}, \tag{9}$$

where $\Lambda$ is the constraint on the average learning cost and accounts for delays in learning. Since the optimization queue can potentially be infinite, to ensure that new optimization tasks at the end get evaluated, we state the following lemma,

**Lemma 1.** *(Queue stability) Let the smallest success probability be $\Upsilon_{\min} = \min_{y^O \in \mathcal{S}_O} \Upsilon(u = 1, y^O)$. If,*

$$\frac{\delta M}{\Upsilon_{\min}} < 1 - \frac{\Lambda}{l(0, W)},$$

*then every policy satisfying the constraint in (9) induces a stable queue and a recurrent Markov chain.*

Lemma 1 ensures that the optimization queue is stable. Since the policy of always transmitting satisfies the constraint, the space of policies that induce a recurrent Markov chain is non-empty.

### 3.3 Structural Results

This subsection proves structural results for the optimal policy solving the MDP and the CMDP. The threshold structure of the optimal policy substantially reduces the search space and is used in devising the structured policy gradient algorithm. The following theorem proves that the optimal policy, $\nu^*$ solving the finite horizon MDP of (7) has a threshold structure with respect to the learner state, $y_n^L$.

**Theorem 2.** *(Nature of optimal policy $\nu^*$) The optimal policy for the finite horizon MDP of (7) with assumptions (M1-3) is deterministic and monotonically increasing in the learner state, $y_n^L$.*

Since the action space consists of 2 actions, a monotonically increasing policy is a threshold policy (Fig. 1(a)),

$$\nu^*(y_n) = \begin{cases} 0 = \texttt{obfuscate}, & y_n^L < \chi(y_n^O) \\ 1 = \texttt{learn}, & \text{otherwise} \end{cases},$$

where $\chi(y_n^O)$ is an oracle state dependent threshold and parametrizes the policy. The proof of Theorem 2 is in the Appendix and follows from Lemma 3, supermodularity, and assumptions on the cost and transition probability matrix.

In order to characterize the structure of the optimal policy for the CMDP, we first study an unconstrained Lagrangian average cost MDP with the instantaneous cost, $w(u, y^O; \lambda) = c(u, y^O) + \lambda l(u, y^O)$, where $\lambda$ is the Lagrange multiplier. The average Lagrangian cost for a policy $\nu$ is then given by,

$$J_{y_0}(\nu, \lambda) = \lim\sup_{N \to \infty} \frac{1}{N} \mathbb{E}_\nu \left[ \sum_{n=1}^{N} w(u_n, y_n^O; \lambda) \mid y_0 \right], \quad \forall y_0 \in \mathcal{S}.$$

The corresponding average Lagrangian cost MDP and optimal stationary policy are,

$$\begin{aligned} J_{y_0}^*(\lambda) &= \inf_{\nu \in \mathcal{T}} J_{y_0}(\nu, \lambda), \\ \nu_\lambda^* &= \arg\inf_{\nu \in \mathcal{T}} J_{y_0}(\nu, \lambda). \end{aligned} \tag{10}$$

Further, we treat the average Lagrangian cost MDP of (10) as a limiting case of the following discounted Lagrangian cost MDP when the discounting factor, $\beta$ goes to 1,

$$J_{y_0}^\beta(\nu, \beta, \lambda) = \lim\sup_{N \to \infty} \mathbb{E}_\nu \left[ \sum_{n=1}^{N} \beta^n w(u_n, y_n^O; \lambda) \mid y_0 \right]. \tag{11}$$

Theorem 2 can then be extended to show that the optimal policy of (11) has a threshold structure in terms of the learner state, $y_n^L$. The average Lagrangian cost MDP of (10) is a limit of the discounted Lagrangian cost MDPs (11) with an appropriate sequence of discount factors $(\beta_n)$ converging to 1, and therefore has a threshold policy. The existence of a stationary policy for (10) is shown in previous work Ngo & Krishnamurthy (2010); Sennott (1989) and is discussed in Appendix B.4. Hence we directly state a corollary from Sennott (1989) as a theorem, which shows that the stationary queuing policy for the CMDP in (9) is a randomized mixture of optimal policies for two average Lagrangian cost MDPs.

**Theorem 3.** *(Existence of randomized stationary policy) (Sennott, 1993) There exists an optimal stationary policy for the CMDP of (9), which is a randomized mixture of optimal policies of two average Lagrangian cost MDPs,*

$$\nu^* = p\nu^*_{\lambda_1} + (1-p)\nu^*_{\lambda_2}, \tag{12}$$

*where $\nu^*_{\lambda_1}$ and $\nu^*_{\lambda_2}$ are optimal policies for average Lagrangian cost MDPs of the form (10) and $p$ is the probability with which $\nu^*_{\lambda_1}$ is chosen.*

Because of Theorem 3, the optimal policy for the CMDP is a randomized mixture of two threshold policies and will also have a threshold structure. A threshold policy of the form of (12) has two threshold levels (denoted by $\phi_1$ for transition from action 0 to randomized action and $\phi_2$ for randomized action to action 1) can be written as,

$$\nu^*(y) = \begin{cases} 0, & y^L < \phi_1 \\ 1 \text{ with prob. } p, & \phi_1 \leq y^L < \phi_2 \\ 1, & \phi_2 \leq y^L \end{cases}. \tag{13}$$

We next propose an efficient reinforcement learning algorithm that exploits this structure to learn the optimal stationary policy for the CMDP.

### 3.4 Structured Policy Gradient Algorithm

The optimal policies for both the finite horizon MDP and infinite horizon MDP using existing techniques of either value iteration or linear programming, but these methods require knowledge of the transition probabilities. Hence to search for the optimal policy without the knowledge of the transition probabilities, we propose a policy gradient algorithm which has a linear time complexity in the learner state space.

In order to efficiently find the optimal policy of the form of (13) we use an approximate sigmoidal policy $\hat{\nu}(y, \Theta)$ constructed as follows,

$$\hat{\nu}(y, \Theta) = \left( \frac{h}{1 + \exp\frac{-y^L + \theta_1}{\tau}} + \frac{1-h}{1 + \exp\frac{-y^L + \theta_2}{\tau}} \right), \tag{14}$$

where $\theta_1$ and $\theta_2$ are a parameter which approximates the thresholds and the randomization factor $p$, $\phi_1$ and $\phi_2$. $h$ approximates the mixing probability $p$. Parameter $\tau$ controls how close the sigmoidal policy follows a discrete threshold policy. It can be shown that as $\tau \to 0$, $h \to p$, $\theta_1 \to \phi_1$ and $\theta_2 \to \phi_2$, the approximate policy converges to the true policy $\hat{\nu}(y, \Theta) \to \nu^*$ (Ngo & Krishnamurthy, 2010; Kushner & Yin, 2003).

---

**Algorithm 1** Structured Policy Gradient Algorithm

**Input:** Initial Policy Parameters $\Theta_0$, Perturbation Parameter $\omega$, $K$ Iterations, Step Size $\kappa$, Scale Parameter $\rho$, Learning cost $l$, Privacy Cost $c$

**Output:** Policy Parameters $\Theta_K$

    **procedure** COMPUTESTATIONARYPOLICY($\omega, K, \kappa, \rho$)
        **for** $n \leftarrow 1 \ldots K$ **do**
            $\Gamma \leftarrow Bernoulli(\frac{1}{2})$                         ▷ $3 \times |\mathcal{S}_O||\mathcal{S}_E|$ i.i.d. Bernoulli random variables
            $\Theta_n^+ \leftarrow \Theta_n + \Gamma * \omega, \Theta_n^- \leftarrow \Theta_n - \Gamma * \omega, \hat{l} \leftarrow \text{AVGCOST}(l, \Theta_n)$
            $\hat{\Delta l} \leftarrow \left(\text{AVGCOST}((l, \Theta_n^+) - \text{AVGCOST}(l, \Theta_n^-)\right), \hat{\Delta c} \leftarrow \left(\text{AVGCOST}(c, \Theta_n^+) - \text{AVGCOST}(c, \Theta_n^-)\right)$
            $\Theta_n$ and $\xi_n$ using (15)
        **end for**
    **end procedure**
    **procedure** AVGCOST($J, \Theta$)
        $\hat{\nu} \leftarrow \text{POLICYFROMPARAMETERS}(\Theta)$
        $\hat{J} \leftarrow \frac{1}{T} \sum_{t=1}^{T} J(\hat{\nu}(y_t), y_t)$
    **end procedure**

---

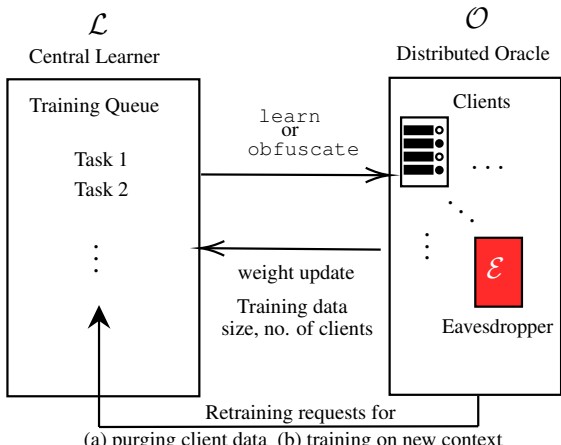

Figure 2: Learner $\mathcal{L}$ either learns from the oracle $\mathcal{O}$ (distributed clients) or obfuscates the eavesdropper $\mathcal{E}$ based on the oracle state (number of clients) and the queue state (number of successful training rounds). Learner (central aggregator in FL) updates its queue state based on the response (size of the training data) and the action taken.

A policy gradient algorithm that finds an optimal policy of the form (12) for solving the CMDP of (9) is now proposed. For each oracle state, the parameters for the approximate policy of (14) is given by $(\theta_1, \theta_2, h)$. The complete set of policy parameters (total of $3 \times |\mathcal{S}_O||\mathcal{S}_E|$ parameters) is referred to as, $\Theta$. The procedure is summarized in Algorithm 1.

Our policy gradient algorithm updates the parameter estimates $\Theta_n$ using (15) by taking an approximate gradient of the average costs, the constraint in (9) and $\xi_n$ which updates using (15). The parameters which are updated are chosen randomly using a vector $\Gamma$, the components of which have an i.i.d. Bernoulli distribution. These chosen parameters are perturbed by $+\omega$ and $-\omega$ and the approximate gradient of the privacy and learning cost is denoted by $\Delta c$ and $\Delta l$, respectively. The approximate gradients of these costs are computed using a finite difference method on the approximate costs. The approximate average costs are computed by interacting with the Markovian oracle for $T$ timesteps. The approximate average learning cost is denoted by $\hat{l}$. $\kappa$ and $\rho$ are suitably chosen step and scale parameters [7].

$$\Theta_{n+1} = \Theta_n - \kappa \left( \Delta c + \hat{\Delta} l \times \max \left[ 0, \xi_n + \rho \left( \hat{l} - \Lambda \right) \right] \right),$$

$$\xi_{n+1} = \max \left[ \left( 1 - \frac{\kappa}{\rho} \xi_n \right), \xi_n + \kappa \left( \hat{l} - \Lambda \right) \right]. \tag{15}$$

The SPGA algorithm can run on a faster time scale by interacting with the system parallel to the SGD and updating the stationary policy parameters, $\Theta$. The stationary policy controls the query posed and updates the SGD estimate in (1). The computational complexity for the SPGA algorithm described here is $O(|\mathcal{S}|)$, i.e., the algorithm is linear in the state space and hence significantly more scalable than standard policy gradient methods (Kushner & Yin, 2003).

### 3.5 Alternative discrete optimization formulation for threshold identification

Since the SPGA algorithm is not amenable to finite time complexity analysis, an alternative approach to solve for the threshold levels is by formulating the problem as multi-armed bandits (MAB). Considering each possible configuration of $Y = |\mathcal{S}_O||\mathcal{S}_E|$ thresholds (each of which can take $\upsilon$ values) for the Lagrange constrained problem of (11) as arms in a MAB problem, the minimum achievable regret is of the order $O\left( \upsilon^Y \log T \right)$ (Lai & Robbins, 1985). The value of the Lagrange parameter can then be iterated over similar to the value iteration of Ngo & Krishnamurthy (2010) to obtain the two Lagrange multipliers and the randomization probability which form the optimal policy of (13). The main disadvantages of a MAB approach compared to the SPGA is that there are strong assumptions on the noise structure, it is unable to track changes in the underlying system and the time complexity is exponential in the state space.

---

[7]The step size is constant if we want the SPGA to track change in the underlying system and decreasing otherwise.

# 4  Hate speech classification in a federated setting

We now present a novel application of the covert optimization framework in designing a robust hate speech classifier illustrated in Figure 2. The task of the learner $\mathcal{L}$ is to minimize a classification loss function ($f(x)$) and simultaneously hide the optimal classifier (neural network) parameters ($\arg\min f(x)$) from an eavesdropper ($\mathcal{E}$).

**State Space:** In our federated setting, the oracle state, $y^O$ denotes the number of clients participating in each communication round, and more clients indicate a better state. Client participation is assumed to be Markovian since it is more general than i.i.d. participation and is closer to a real-world scenario (Sun et al., 2018; Rodio et al., 2023). Although the stochastic aspect of the oracle is modeled on client participation and the quantity of the training data, it can also be modeled with respect to the quality. For example, in hate speech detection and similar applications, unintended bias based on characteristics like race, gender, etc. often occurs (Dixon et al., 2018). If the oracle is based on how diverse the training data is, we can train when the available data is good enough and obfuscate otherwise using costs related to biased classification (Viswanath et al., 2022). The learner state, $y^L$, denotes the number of remaining successful gradient updates. The learner decides the total number of required successful gradient updates based on convergence criteria, like the one in Theorem 1. A successful gradient update is done if the number of available training data is above a threshold, similar to communication skipping using system parameters in Sun et al. (2022); Mishchenko et al. (2022). This is a proxy for the threshold of $\sigma$ in (A2) since an exact noise bound is difficult to obtain in practice. The optimization queue receives new arrivals, $y^E$ for model retraining due to client requests for unlearning their data, data distribution shifts, and active learning (Bachman et al., 2017; Sekhari et al., 2021; Cai et al., 2021). The timescale for practical federated training ranges from a few hours to a few days (Hard et al., 2019). In the hate speech classification, model retraining requests due to a shift in context is on a slower time scale, but purging requests for a client's data can be made every few hours. All devices, including the ones not participating in the training round can make such a request ensuring $y^E$ is independent of $y^O$ (E1).

**Action Space:** Using labeled datasets, GANs can be trained to generate hate speech (Lin et al., 2017; de Masson d' Autume et al., 2019). But since access to labeled data is difficult, an eavesdropper $\mathcal{E}$ can use the optimal weights as discriminator weights to train a generator (Wang et al., 2018). The action space, $\mathcal{U}$ is to either send the correct learning neural network parameters or the obfuscating parameters. We discuss in the next section how to generate obfuscating parameters under two different eavesdroppers' information scenarios.

**Costs:** Although for this paper, only a majority of queries need to be obfuscating, in general, the more learning queries an eavesdropper knows, the higher probability of the eavesdropper figuring out the optimal weights. Hence the eavesdropper can generate hate speech and misinformation which are semantically more coherent (measured by metrics of readability and perplexity (Carrasco-Farré, 2022; Mimno et al., 2011)) and can be spread easily (Viswanath et al., 2022). Hence the privacy cost in the MDP can be associated with metrics of the spread of malicious content, including prevalance (Wang et al., 2021), and contagion spread rates (Davani et al., 2021; Lawson et al., 2023). Analogously, the learning cost $l$ can be associated with the same set of costs since malicious content will go unclassified if the classifier does not achieve the desired accuracy. For delays in forgetting a client's information, the learning cost is analogous to the value of private information, measured by metrics like Shapeley value (Kleinberg, 2001; Wang et al., 2020). Also, clients might want to remove their data because they incorrectly annotated it earlier, and keeping the retraining task in the queue can worsen the real-time accuracy (Davani et al., 2021; Klie et al., 2023). The privacy and learning cost and the learning constraint, $\Lambda$ of (9), can be chosen appropriately depending on the maximum ratio of the queries the learner can afford to expose, which for the proportional sampling estimator is 0.5.

## 4.1  Numerical Results

For the first experiment, the MDP formulation and structural results are demonstrated on a hate speech classification task under a federated setting described above. The convergence of the threshold parameters in SPGA is investigated in the second experiment. It is empirically shown that the optimal policy solving (9) is threshold in the oracle state, demonstrating that the optimal policy makes the learner learns more when the oracle is good. Additional benchmark experiments on the MNIST data, dataset preprocessing, the architecture, and assumptions are listed in the Appendix. Before discussing the numerical study, we explain two scenarios with respect to the information that the eavesdropper has and the information the learner has about the eavesdropper. The results of the study are then presented under both scenarios.

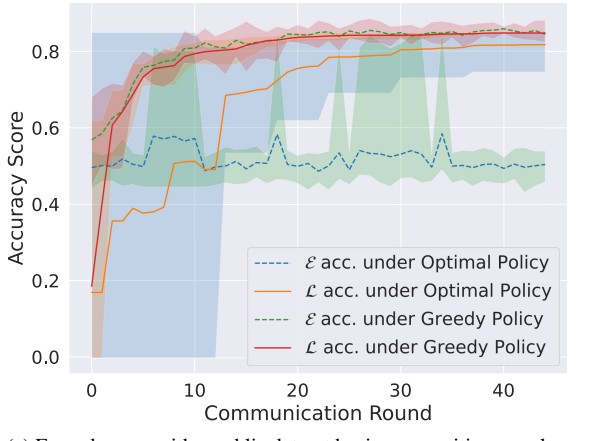 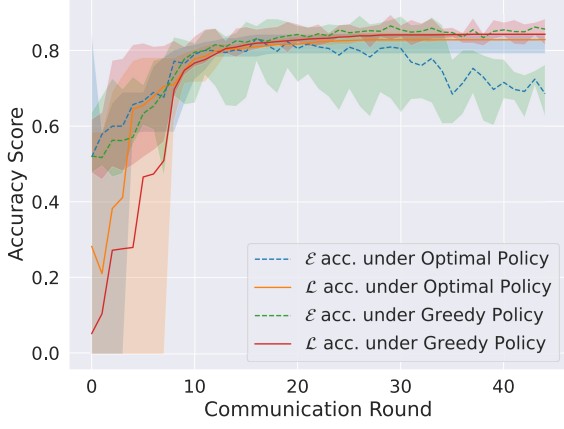

(a) Eavesdropper with a public dataset having no positive samples.  (b) Eavesdropper with a dataset having 10% positive samples.

Figure 3: Convergence of validation accuracies for Learner $\mathcal{L}$ and Eavesdropper $\mathcal{E}$ under greedy and optimal policy. The optimal policy helps dynamically learn and hide the optimal weights compared to a greedy policy of always learning.

**Scenario 1: Eavesdropper does not have enough data:** When the eavesdropper does not have enough data to choose between the two SGDs, the obfuscating queries can be posed in various ways, for example, by posing noisy queries sampled randomly from domain $\mathbb{R}^d$ or by doing a mirrored gradient descent using the true queries in (5). Since the eavesdropper has no information, the prior over the two SGDs is the same. An extension of this scenario is when the learner is trying to obfuscate hyperparameters which are essential to training a neural network (Probst et al., 2019) from the eavesdropper. The learner can switch between the intended hyperparameter and a suitably chosen obfuscating hyperparameter.

**Scenario 2: Eavesdropper has a subset of data, but the learner has information about the subset:** In case the eavesdropper has access to a subset of the data, it can test out both the trajectories on its dataset to see which one has a smaller empirical loss. Let the dataset of the eavesdropper be $D_0 \in D$. If the learner knows $D_0$, then it can simulate an oracle with function $f'(x) = f(x, D_0) = \frac{1}{|D_0|} \sum_{d \in D_0} \mathcal{G}(x, d)$ to obtain noisy gradients, $r'_k = \nabla f'(q_k) + \eta'_k$ where $\mathcal{G}$ is the loss function and $\eta'_k$ is suitably simulated noise. The obfuscating queries can be obtained using the following SGD trajectory: $\hat{z}_k = \hat{z}_{k-1} - \mu_k r'_k \mathbb{1}(u_k = 0)$. For example, when the eavesdropper is using a public dataset, accessing a reliable and balanced dataset is otherwise costly. The case where learner has incomplete information about the eavesdropper's dataset is left for future research. The case when the eavesdropper has access to all data is not relevant since then the eavesdropper can carry out the optimization on its own. The weights obtained using the parallel SGD do not generalize well since the empirical loss being minimized is for a subset of the data. This makes the prior over both the SGDs more balanced since the eavesdropper can no longer take advantage of its dataset. And because the parallel SGD is observed the majority of the time, then the eavesdropper's estimate (6) corresponds to the parallel SGD.

## 4.2 Demonstration of MDP framework on Covert Optimization in Hate Speech classification

A hate speech classification task is considered where an extended version of a pre-trained BERT (Devlin et al., 2019; Turc et al., 2019) model is fine-trained on a labeled dataset to classify textual data as toxic or not [8]. A federated setting with 20 clients whose data is non-overlapping is considered. Each client has $5443$ training samples and $1443$ validation samples. For the experimental results, we consider $N = 45$ communication rounds (or queries) and $M = 16$ successful model updates (which is around $\sim 34\%$ of the total queries). To demonstrate the versatility of our formulation, we use the FedAvg algorithm, where the learner aggregates the weights of the individual clients rather than the weight updates (McMahan et al., 2017). The experiment is done with $10$ random seeds to obtain error bounds. We use size of the training data as a proxy for $\sigma_k$ since an exact $\sigma_k$ is difficult to estimate. A threshold on the noise bound ($\sigma$) is approximated with the threshold on the number of data points, i.e., the learner discards any communication round when

---

[8]The dataset used was made public by Jigsaw AI and can be found here. Hate speech classification is still an open problem and the achieved accuracy is barely satisfactory but our aim was to show the application of our formulation. Our source code can be found on this anonymized link.

less than $1/4$ of the entire dataset is available for training[9]. The underlying Markov chain of the device participation has three states, i.e. either $W_1 = 1/4$, $W_2 = 1/2$ or $W_3 = 1/1$ of the devices participating in any communication round. Each device can contribute any number of data points out of the available datapoints, and for the chosen criteria, the success function empirically comes out to be $\Upsilon(y^O, 1) = [0.1, 0.43, 0.95]$. The transition probabilities between the oracle states is given by $P^O = [0.8\ 0.2\ 0; 0.3\ 0.5\ 0.2; 0\ 0.2\ 0.8]$. The privacy cost, $c$ is taken to be $0.3, 0.8, 1.8$ for the respective oracle states and the learning cost, $l$ to be $0.6$. The choice of the cost is done by interacting with the system and seeing the average proportion of queries obfuscated using the stationary policy as shown in Appendix A.4.3. This cost can also be tuned in an online fashion based on the number of slots available, the learner's preference, and the percentage of queries the learner can afford to expose hence enabling practical realizability of previous work on covert optimization (Xu et al., 2021a). Since this is a finite horizon MDP, the learner can use linear programming or value iteration to find an optimal policy using empirical estimates of the transition probabilities using past data or can use the SPGA algorithm and interact with the system to find a stationary sub-optimal policy. We show our results using a stationary policy obtained by using the SPGA algorithm, hence the learner does not know $\mathbb{P}(y_{n+1}|y_n, u_n)$.

Figure 3(a) shows the convergence of the aggregated validation accuracies for the learner and the eavesdropper under Scenario 1 under a greedy and stationary policy. The loss function considered is the binary cross entropy loss function, and the accuracy is the validation accuracy score. The eavesdropper accuracy is calculated using a balanced validation dataset of size 2886. It can be seen that although the learner's accuracy, on average goes up to $0.85$, the eavesdropper's accuracy goes up to $0.52$. Figure 3(b) is for Scenario 2 with an eavesdropper with an imbalanced dataset of $10\%$ toxic (one of the two classes) examples. This dataset is assumed to be public, and the learner has complete access to it, which it uses to obfuscate the eavesdropper. In Figure 3(b), it is evident that the obfuscation achieved is lesser than when the eavesdropper had no toxic samples since the eavesdropper is able to achieve an accuracy of around $0.69$, explained by the fact that the obfuscating parameters are trained on a sample of the entire dataset. Although, in both cases, when the learner uses a greedy policy, the eavesdropper's accuracy is at par with the learner's accuracy. This demonstrates how using the optimal policy, the learner can prevent an eavesdropper from learning the optimal weights of the classifier.

### 4.2.1 Convergence of SPGA algorithm and numerical structural result

The convergence of Algorithm 1 to the true threshold parameters is investigated next. In addition to the previously defined parameters, a learning constraint $\Lambda = 0.2$ is imposed on the average learning rates, setting up a CMDP whose objective is given by (9). The arrival probability for $M = 4$ queries is $\delta = 0.1$ [10]. For calculating the approximate average cost, we take a sample path of $100$ timesteps. The results are averaged over $100$ runs. The convergence of the threshold parameter $\phi_2$ for different oracle states with arrival state $E = 0$ is plotted in Figure 4 along with the true threshold parameters found by linear programming.

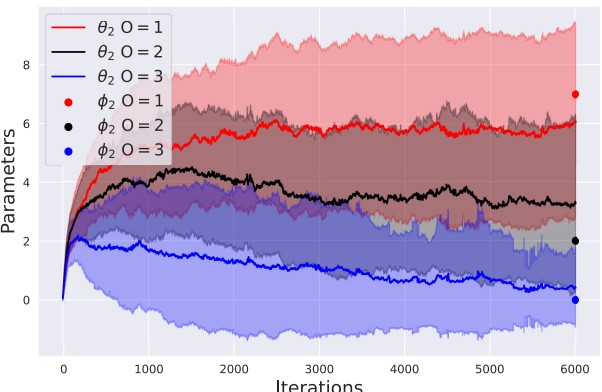

The approximate thresholds, $\theta_2$ of the sigmoidal policy of (14) converge close to the true threshold parameters, $\phi_2$ without the knowledge of the transition probabilities. The threshold for the oracle state 3 converges to a negative value which when plugged into the sigmoidal policy of (14) resemble a threshold policy with threshold at 0 since the learner state can not be negative. It can be numerically

Figure 4: Convergence of threshold parameters $\theta$ to true threshold parameters $\phi$ in Alg. 1 for different oracle states. The optimal policy incentivizes learning more when the oracle is in a better state.

seen that the threshold of optimal policy decreases with increasing oracle state; that is, the optimal policy is non-increasing in the oracle state hence the learner poses a learning query more often when the oracle is in a good state.

---

[9]Recent work in federated learning has proposed skipping training rounds when the distributed oracle is not good enough leading to less communication rounds and better convergence rates (Chen et al., 2018; Sun et al., 2022; Mishchenko et al., 2022).

[10]This is slightly different from the theoretical model since it is not possible to simulate an infinite buffer, we consider a length 40 queue, and to prevent a queue overflow a high learning cost of 100 is imposed in case the queue full. This consideration is similar to previous work on network queueing using a CMDP approach and does not change the threshold and monotone nature of the policy (Djonin & Krishnamurthy, 2007).

The parameters for the SPGA algorithm along with an additional experiment with constant step size is given in Appendix A.6.

## 5 Conclusion

The problem of covert optimization is studied from a dynamic perspective when the oracle is stochastic, and the learner receives new optimization requests. The problem is modeled as a Markov decision process, and structural results are established for the optimal policy. A linear time policy gradient algorithm is proposed, and the application of our framework is demonstrated in a hate speech classification context. Future work can look at inverse RL techniques for the eavesdropper to infer the optimal learner policy, and more robust obfuscating gradient trajectories can be studied. The problem of covert optimization can be investigated in a decentralized setting where the problem is modeled as a switching control game with participants switching between learning and obfuscating others. Our suggested methodology can dynamically control learning in distributed settings for objectives like energy efficiency, client privacy, and client selection. An eavesdropper with finite memory and an objective of minimizing average learning cost with a constraint on average privacy cost can be considered. This work's main broader ethical concerns are a) an increase in energy consumption to achieve covertness and b) it could help a learner covertly train a classifier for questionable reasons, e.g., censorship.

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

# A    Appendix A: Experimental Parameters and Methodology

## A.1    Dataset and Preprocessing

We use Jigsaw's *Unintended Bias in Toxicity Classification* dataset for our experimental results. The dataset has $\sim 1.8$ million public comments from the Civil Comments platform. The dataset was annotated by human raters for toxic conversational attributes, mainly rating the toxicity of each text on a scale of 0 to 1 and sub-categorizing for severe toxicity, obscene, threat, insult, identity attacks, and sexually explicit content. More information about the annotation process can be found on the Kaggle website for this dataset here. We consider the much simpler task of classifying the text as toxic or not. The toxicity is scored by human volunteers on a scale of 0 to 1, and we consider a comment toxic if the toxicity score is greater than $0.5$. The original dataset is imbalanced with $1660540$ non-toxic samples and $144334$ toxic samples, and for each experimental run, we take a random balanced subset with $144334$ toxic and non-toxic samples. The reason we do this is twofold, a) to reduce the time it takes to train our model and make it feasible to run the clients concurrently on our machine, and b) to study the accuracy that the eavesdropper achieves on the positive samples more profoundly. To achieve b), we could also have taken a weighted accuracy function. We preprocess the data by removing special characters and contracting the word space (for example, replacing "can't" and "cant" with the same word).

## A.2    Architecuture, training hyperparameters, and loss functions

Our architecture used for training involves the following layer sequence: A pre-trained BERT layer which outputs a 128-length embedding, a fully connected 128 neurons wide linear layer with ReLU activation, a dropout layer with a rate of $10^{-1}$ and finally, a linear layer classifying the text as hate speech or not. The motivating reason for choosing this architecture was that this was the standard template in many of the submissions received in the competition. However, as highlighted before, our approach is both architecture and convergence algorithm-agnostic. We consider the logit loss function. We use the following hyperparameters for training: learning rate: $10^{-3}$, training batch size of $40$, and validation batch size of $20$. To demonstrate the versatility of our methods, we optimize our neural network using Adam (Kingma & Ba, 2017) and run FedAvg (McMahan et al., 2017) instead of FedSGD. Using the preprocessed training data, we fine-train our model to minimize the binary cross entropy loss (BCE).

### A.3 Markov decision process parameters

We consider 20 clients and client participation is simulated using a Markov chain with the states being 5 clients, 10 clients, and 20 clients. In each training round, each participating client chooses between 0 to $N_{\text{batches}}$ batches. We perform $N = 45$ training rounds with $M = 20$ successful updates. The accuracies are accuracy scores evaluated on the validation dataset averaged across clients. At any given communication round, we assume that the eavesdropper takes whatever weight trajectory occupies the majority number of communication rounds up till that round.

### A.4 Additional benchmark experiments on MNIST dataset

We further conduct experiments on an image recognition task on the MNIST data in a federated setting to a) benchmark our methods on a standard dataset and b) study the effect of varying eavesdropper and Markov chain parameters on the effectiveness of our approach. Also, the image recognition task is more computationally efficient than the hate speech classification task. We can perform a lot more runs of the experiment (20 runs of the Hate Speech Classification task took around $\sim$ 23 hours, whereas, within the same time frame, we could do 1040 runs of the image classification task).

#### A.4.1 Varying eavesdropper parameters

We use a Markov chain which is identical to Experiment 1. We set $M = 30$ successful gradient updates out of 120 communication rounds for this task. We report our results for 20 and 50 clients.

Our experimental results are reported in Table 4 and Table 5 and we summarize our key findings below:

- We vary the size of the training data the eavesdropper has compared to the size of a participating client. We consider three cases: the size of the eavesdropper's training set is $10\%, 40\%$, or $100\%$ of the size of the participating client's size. We observe that as the training size even with the obfuscated weights, the eavesdropper's accuracy improves since the eavesdropper can learn on its own just well enough.

- We also vary the number of classes the eavesdropper has more samples of and the proportion of data for these classes to other classes. We consider three cases: $2, 5$, and $8$ "good" classes that the eavesdropper composes $99\%$ or $90\%$ of the data. The case when all classes are evenly distributed is also benchmarked against. We observe that both the number of good classes and a more balanced dataset improve the eavesdropper accuracy.

- We conclude that for the case of a limited information eavesdropper, the optimal policy performs much better than a greedy policy with the eavesdropper accuracy having a difference of as much as $51\%$ (for the case with 2 good classes which form $99\%$ of the training data with $40\%$ of the size.).

#### A.4.2 Results on FedSGD

For completeness, we demonstrate our results on FedSGD as well with 75 successful updates in 240 communication queries. We summarize our results in Table 3 averaged for 20 experiment runs. The eavesdropper is assumed to have 4 good classes composing $90\%$ of the data and $10\%$ of the training data size. We see that the pattern is similar to the FedAvg case but with slower convergence, with the average learner accuracy going up to $88\%$ while the eavesdropper is stalled up to $56\%$ while the learner uses the optimal policy.

#### A.4.3 Varying Markov chain parameters

| | FedSGD | | 7 oracle states | |
|---|---|---|---|---|
| Type of Policy | Eavesdropper accuracy | Learner accuracy | Eavesdropper accuracy | Learner accuracy |
| Greedy | 0.89 | 0.89 | 0.93 | 0.88 |
| Optimal | 0.56 | 0.88 | 0.34 | 0.88 |

Table 3: Additional results on MNIST dataset using a) FedSGD and b) 7 oracle states demonstrate versatality of our framework and robustness to the choice of system parameters respectively.

We consider a setup similar to experiment 1 with 100 clients and fix the eavesdropper parameters to have 2 good classes composing 99% of the data and 100% of the training data size. The number of oracle states is increased to 7 with the device counts as $[36, 41, 45, 50, 55, 58, 100]$ and the threshold is set to $1/8^{\text{th}}$ of the dataset. The emperical success probabilities are found to be $[0.01, 0.12, 0.41, 0.75, 0.94, 0.96, 1]$. The probability transition matrix has a structure similar to the previous one and can be found in the code.

The results are summarized in Table 3 averaged over 30 runs, and we see trends similar to the case with 3 oracle states (the eavesdropper accuracy is much less since the training data is now distributed over 100 clients so the eavesdropper data is not big enough) . The possible reason for the eavesdropper having a better accuracy in a greedy scheme could be that the weights that the eavesdropper estimates are optimal and the eavesdropper has more time slots to train.

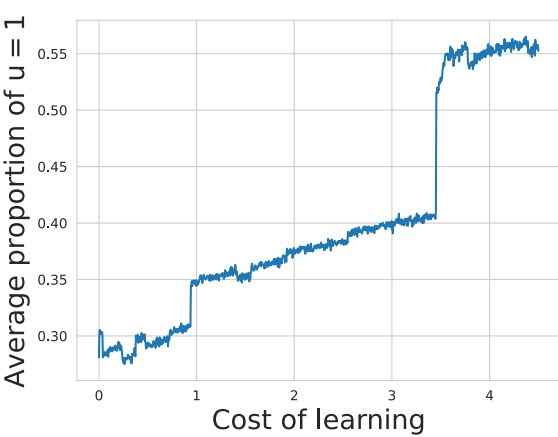

Figure 5: Empirical results on the effect of the cost of learning on the average proportion of learning queries.

We also investigate the effect of learning cost, $l$, on the average number of times action $u = 1$ (learn) is taken when using the stationary policy in Fig. 5. This helps illustrate how our learning cost can be chosen to achieve a desired level of learner-privacy (percentage of queries that should be obfuscated), which are set based on theoretical results (Xu et al., 2021b). The average cost is calculated over 1000 for different simulation runs. The graph seems piecewise linear with jumps which could be explained by the nature of the solution of the occupation measure for the MDP, which is a vertex of a polytope defined by the corresponding linear program and jumps to next vertex with an increase in the instantenous cost.

## A.5 Discussion on a policy without any stochastic considerations and possible limitations of the optimal policy

In the controlled SGD setting presented in this paper, the learner discards certain queries which are evaluated unsuccessfully according to a chosen noise constant. Controlling for stochasticity helps filter out communication rounds that will perform poorly due to insufficient clients and training data. When there is an appreciable disparity between the good and bad states, such filtering leads to a significant performance difference. Additionally, our assumption is more general than a single noise constant. The learner can control the noise threshold constant, $\sigma$, which enables the characterization of the finite number of updates to optimize $f$.

An approach without such considerations is different, it will be equivalent to using a random policy in placing $M$ learning queries in $N$ total queries. Our approach can solve this task by modifying the SGD update step and updating the learner state regardless of $\sigma_k$. For a practical setting, such a scheme would still perform better than the random policy since the obfuscation would be the same, but the learning would be done when the practical parameters, like the number of clients and data points, are better. In our numerical experiments with the MNIST data, using a random policy of placing 30 learning queries in 120 communication rounds, the learner obtained a average accuracy of $80.5\%$ against an accuracy $92.5\%$ obtained using the optimal policy.

The learning performance of the proposed optimal policy is the same as the greedy policy since the greedy policy also uses the same thresholding ($\sigma_k < \sigma$) policy. However, since the greedy method does not schedule the queries optimally, it poses a lot more learning queries and cannot achieve obfuscation as illustrated in the numerical experiments. On the other hand, a random policy without thresholding, which poses M learning queries in N communication slots, achieves obfuscation but suffers in learning performance. Hence, the optimal policy can be seen as achieving a tradeoff between both of them. The significant limitations of using the proposed method are:

1. The gain from the optimal policy will not be significantly high if the ratio M/N is much higher than the average occupancy of the oracle in the good oracle states.

2. The stochastic nature of the optimal policy implies that there is a probability that enough queries are not obfuscated.

## A.6  SPGA algorithm parameters and additional experiment

For analysing the convergence of the SPGA algorithm, we choose the step size $\kappa_n = \frac{0.5}{n}$, the scale parameter as $\rho = 20$ and the initial constraint parameter as $\xi = 10$. The initial condition for the learner state is set to be $y^L = 40$ and the oracle is state $y^O = 3$ when interacting with the system.

In Fig. 7, we also demonstrate how with a constant step size the SPGA algorithm is able to track changes in the underlying policy parameters. Before iteration 2000 the underlying Markov chain has parameters same as the previous SPGA experiment except the arrival rate which is $3\%$ for $M = 10$ updates. After the 2000 iteration, this changes to $M = 4$ updates, with an arrival rate of $10\%$. The success probabilities and the oracle state transition is taken to be, $\Upsilon = [0.1, 0.6, 0.9]$ and $P^O = [0.7\ 0.2\ 0.1; 0.3\ 0.1\ 0.7; 0.2\ 0.2\ 0.7]$. The results are averaged over 100 runs. It can be seen from the figure that even though the convergence is not as close as the decreasing step size, the SPGA algorithm tracks changes in the system. This effect is more prominently visible for oracle state $y^O = 1$ since the change in the true parameters is significant.

**Note on convergence:** Proving convergence for the SPGA algorithm is difficult since policy search is not a convex problem and it's difficult to make regularity assumptions for the average cost function of the CMDP with respect to the policy (which is on a discrete space). Efficient linear programs can be derived for solving the CMDP which have finite sample results but these methods require knowledge of the transition probabilities (Mattila et al., 2017). Hence we use the structural results on the optimal policy to reduce the search space from $|\mathcal{S}_{\mathcal{O}}||\mathcal{S}_L||\mathcal{S}_E|$ to $3|\mathcal{S}_{\mathcal{O}}||\mathcal{S}_E|$. Where $|\mathcal{S}_L|$ is the finite approximation for the countably infinite queue length. This reduction at least makes it computationally possible to minimize the cost function.

### A.7 Tradeoff between convergence and obfuscation

For a finite-horizon case, the learners convergence varies as $O(\frac{\gamma\sigma^2}{\log m})$ where $m$ is the number of successful updates already made. The obfuscation with respect to an eavesdropper using proportional sampling and MAP estimator is a step function as shown in Figure 6. As a reminder this simple model of eavesdropper works in our setting because the eavesdropper is assumed to be have equal priors on both the trajectories in absence of any additional information.

Existing theoretical results in learner-private optimization give more general probability bounds but till now have been extended to the convex optimization case only (using a Dirichlet process prior) (Xu et al., 2023). Without going into the technical details, the key idea of the bound is that in order to ensure that the eavesdropper has a less than $1/\varrho$ probability of estimating the $\arg\min$, the learner must pose at least $\varrho$ times the standard number of learning queries, where $\varrho \in \mathbb{N}$. Although tight upper and lower bounds are difficult to obtain for non-convex functions even with regularity assumptions, the lower bounds for the convex case serve as loose lower bound for the non-convex case too, making the obfuscation probability a staircase function.

For the infinite horizon case the obfuscation has a similar structure but we look at it with respect to the average number of learning queries and obfuscation on average. And if the conditions of Lemma 1 are

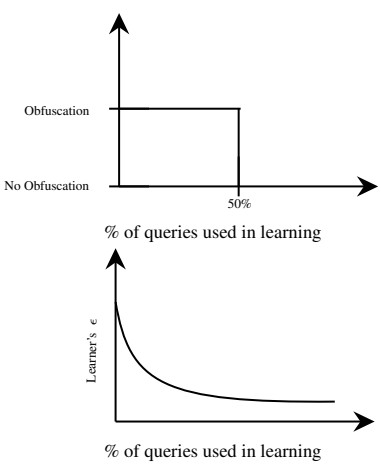

Figure 6: Pictorial representation of the tradeoff between the learning objective and the obfuscation objective. (a) shows obfuscation is a step function for the proportional sampling estimator based eavesdropper. (b) shows the change in closeness to a critical point of $f$.

satisfied, the optimization remains stable and the set of optimization tasks converge and the learning objective is satisfied. Note that for the infinite horizon case the underlying true emperical function $f$ changes with a change in context or removal of a data point. Our methods could also be extended to an SGD which has a constant step size in which the asymptotic convergence rate becomes slower as the average number of queries decreases although an exact characterization is tough.

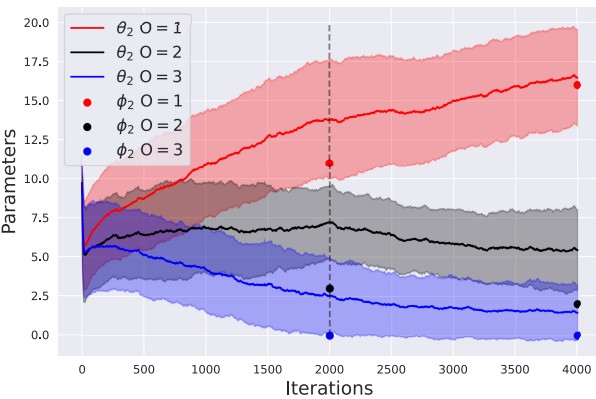

Figure 7: Convergence of threshold parameters in Alg. 1 for different oracle states with constant step sizes. This trajectory for the parameter estimates is more erroneous, has only weak convergence results but can track changes in the underlying true parameters.

### A.8 Note on Arrival State Space

The arrival state space taken in the paper is a specific case for illustrative purposes. It could be a more general set, for ex. $\mathcal{S}^E = \{M, 2M, 3M, 4M\}$ with a suitable probability distribution. Lemma 1 can be suitably adjusted to ensure a stable queue as long as the expected arrival rate is finite. We can then quantize the actual arrivals to the nearest arrival

| | $\mathcal{E}$ Dataset Parameters | | $\mathcal{E}$ Accuracy | | $\mathcal{L}$ Accuracy | |
| No. of Good Classes | Prop. of Training Data | Prop. of Good Classes | Greedy Policy | Optimal Policy | Greedy Policy | Optimal Policy |
|---|---|---|---|---|---|---|
| 2 | 0.1 | 0.99 | 0.857 | 0.336 | 0.925 | 0.832 |
| 2 | 0.1 | 0.9 | 0.857 | 0.725 | 0.925 | 0.832 |
| 2 | 0.1 | 0.2 | 0.857 | 0.927 | 0.925 | 0.832 |
| 2 | 0.4 | 0.99 | 0.859 | 0.445 | 0.924 | 0.815 |
| 2 | 0.4 | 0.9 | 0.859 | 0.907 | 0.924 | 0.815 |
| 2 | 0.4 | 0.2 | 0.859 | 0.973 | 0.924 | 0.815 |
| 2 | 1 | 0.99 | 0.850 | 0.632 | 0.930 | 0.822 |
| 2 | 1 | 0.9 | 0.850 | 0.916 | 0.930 | 0.822 |
| 2 | 1 | 0.2 | 0.850 | 0.968 | 0.930 | 0.822 |
| 5 | 0.1 | 0.99 | 0.843 | 0.784 | 0.924 | 0.817 |
| 5 | 0.1 | 0.9 | 0.843 | 0.900 | 0.924 | 0.817 |
| 5 | 0.1 | 0.5 | 0.843 | 0.932 | 0.924 | 0.817 |
| 5 | 0.4 | 0.99 | 0.834 | 0.707 | 0.922 | 0.819 |
| 5 | 0.4 | 0.9 | 0.834 | 0.920 | 0.922 | 0.819 |
| 5 | 0.4 | 0.5 | 0.834 | 0.970 | 0.922 | 0.819 |
| 5 | 1 | 0.99 | 0.848 | 0.820 | 0.926 | 0.815 |
| 5 | 1 | 0.9 | 0.848 | 0.961 | 0.926 | 0.815 |
| 5 | 1 | 0.5 | 0.848 | 0.977 | 0.926 | 0.815 |
| 8 | 0.1 | 0.99 | 0.830 | 0.898 | 0.926 | 0.819 |
| 8 | 0.1 | 0.9 | 0.830 | 0.957 | 0.926 | 0.819 |
| 8 | 0.1 | 0.8 | 0.830 | 0.943 | 0.926 | 0.819 |
| 8 | 0.4 | 0.99 | 0.868 | 0.898 | 0.926 | 0.809 |
| 8 | 0.4 | 0.9 | 0.868 | 0.936 | 0.926 | 0.809 |
| 8 | 0.4 | 0.8 | 0.868 | 0.955 | 0.926 | 0.809 |
| 8 | 1 | 0.99 | 0.864 | 0.943 | 0.931 | 0.831 |
| 8 | 1 | 0.9 | 0.864 | 0.975 | 0.931 | 0.831 |
| 8 | 1 | 0.8 | 0.864 | 0.975 | 0.931 | 0.831 |

Table 4: Additional experiments on the MNIST data with 20 clients showcase how varying different eavesdropper parameter changes the accuracy the eavesdropper is able to achieve.

state. In most practical implementations of a stochastic gradient algorithm, apart from an epsilon-based convergence criteria, there is also a parameter for the maximum number of iterations, which can be thought of as . Additionally. the implementation for our particular arrival state space $\{0, M\}$ can be done by considering two states (an arrival or no arrival), and the structured policy gradient algorithm does not need the exact value of $M$ to update the state parameters. The SPGA also adapts to changes in the underlying arrival probability.

# B    Appendix B: Proofs

## B.1    Proof of Theorem 1

*Proof.* Let the learner successfully update the minima estimate at time indices, $k_1, \ldots, k_m$. Using quadratic bound associated with the Lipschitz condition of A1 (the descent lemma) the following can be derived. We use a constant step size $\mu$ but similar results can be derived for a decreasing step size.

**Lemma 2.** *(Modified Descent Lemma) For an oracle function $f$ which satisfies assumptions A1, the following can be shown for the successful updates (Def. 1) performed at time $k_1, \ldots, k_M$,*

$$\left(\mu - \frac{\mu^2\gamma}{2}\right) \sum_{m=1}^{M} \|\nabla f(x_{k_m})\|^2 \le f(x_{k_0}) - f^* - \left(\mu - \mu^2\gamma\right) \sum_{m=1}^{M} \langle \nabla f(x_{k_m}), \eta_{k_m}\rangle + \frac{\mu^2\gamma}{2} \sum_{m=1}^{M} \|\eta_{k_m}\|^2. \quad (16)$$

We introduce the following expression to bound the expected gradient norm and obtain the desired result,

$$\Psi_M = \frac{1}{M} \sum_{m=0}^{M-1} \mathbb{E}\|\nabla f(x_{k_m})\|^2.$$

The expectation is with respect to the noise terms $\eta_{k_0}, \eta_{k_1}, \ldots, \eta_{k_M}$. $\Psi_M$ is equal to the expected gradient norm of a uniformly at random selected iterate, $\Psi_M = \mathbb{E}\|\nabla f(\hat{x})\|^2$ for $\hat{x} \in_{\text{u.a.r.}} \{x_{k_1}, \ldots, x_{k_M}\}$. The expectation now is w.r.t.

| | $\mathcal{E}$ Dataset Parameters | | $\mathcal{E}$ Accuracy | | $\mathcal{L}$ Accuracy | |
|---|---|---|---|---|---|---|
| No. of Good Classes | Prop. of Training Data | Prop. of Good Classes | Greedy Policy | Optimal Policy | Greedy Policy | Optimal Policy |
| 2 | 0.1 | 0.99 | 0.861 | 0.320 | 0.860 | 0.840 |
| 2 | 0.1 | 0.9 | 0.861 | 0.611 | 0.860 | 0.840 |
| 2 | 0.1 | 0.2 | 0.861 | 0.852 | 0.860 | 0.840 |
| 2 | 0.4 | 0.99 | 0.884 | 0.318 | 0.862 | 0.830 |
| 2 | 0.4 | 0.9 | 0.884 | 0.734 | 0.862 | 0.830 |
| 2 | 0.4 | 0.2 | 0.884 | 0.895 | 0.862 | 0.830 |
| 2 | 1 | 0.99 | 0.852 | 0.393 | 0.859 | 0.843 |
| 2 | 1 | 0.9 | 0.852 | 0.864 | 0.859 | 0.843 |
| 2 | 1 | 0.2 | 0.852 | 0.955 | 0.859 | 0.843 |
| 5 | 0.1 | 0.99 | 0.875 | 0.461 | 0.638 | 0.838 |
| 5 | 0.1 | 0.9 | 0.875 | 0.655 | 0.638 | 0.838 |
| 5 | 0.1 | 0.5 | 0.875 | 0.807 | 0.638 | 0.838 |
| 5 | 0.4 | 0.99 | 0.843 | 0.595 | 0.854 | 0.833 |
| 5 | 0.4 | 0.9 | 0.843 | 0.795 | 0.854 | 0.833 |
| 5 | 0.4 | 0.5 | 0.843 | 0.916 | 0.854 | 0.833 |
| 5 | 1 | 0.99 | 0.864 | 0.710 | 0.857 | 0.840 |
| 5 | 1 | 0.9 | 0.864 | 0.810 | 0.857 | 0.840 |
| 5 | 1 | 0.5 | 0.864 | 0.892 | 0.857 | 0.840 |
| 8 | 0.1 | 0.99 | 0.868 | 0.408 | 0.856 | 0.839 |
| 8 | 0.1 | 0.9 | 0.868 | 0.714 | 0.856 | 0.839 |
| 8 | 0.1 | 0.8 | 0.868 | 0.870 | 0.856 | 0.839 |
| 8 | 0.4 | 0.99 | 0.841 | 0.805 | 0.862 | 0.832 |
| 8 | 0.4 | 0.9 | 0.841 | 0.916 | 0.862 | 0.832 |
| 8 | 0.4 | 0.8 | 0.841 | 0.914 | 0.862 | 0.832 |
| 8 | 1 | 0.99 | 0.861 | 0.820 | 0.853 | 0.840 |
| 8 | 1 | 0.9 | 0.861 | 0.860 | 0.853 | 0.840 |
| 8 | 1 | 0.8 | 0.861 | 0.881 | 0.853 | 0.840 |

Table 5: Additional experiments on the MNIST data with 50 clients showcase how varying different eavesdropper parameter changes the accuracy the eavesdropper is able to achieve and how our framework can be extended to more number of clients.

both the random iterate and the noise terms. The bound on $\Psi_M$ is presented in the following corollary and is obtained by algebraic manipulation of Lemma 2, taking expectation on both sides and using assumption (A1, A2),

**Corollary 1.** *After $M$ successful updates of the SGD (Def. 1), for any step size $\mu \leq \frac{1}{\gamma}$, under assumptions (A1, A2) the following holds,*

$$\Psi_M \leq \frac{2\left(\mathbb{E}f(x_{k_1}) - f^*\right)}{M\mu} + \mu\gamma\sigma^2. \tag{17}$$

What remains to be shown is that for appropriate step size $\mu$ and number of successful updates $M$, $\Psi_M \leq O(\epsilon)$ and the learning objective is achieved. To achieve this, we obtain the following conditions on the two summands of (17) by bounding them individually by $\epsilon/2$ and setting $F = (\mathbb{E}f(x_{k_1}) - f^*)$,

$$\frac{2F}{M\mu} \leq \frac{\epsilon}{2}, \mu\gamma\sigma^2 \leq \frac{\epsilon}{2}.$$

The second equations bounds the step size, $\mu \leq \frac{\epsilon}{2\gamma\sigma^2}$, which along with $\mu \leq \frac{1}{\gamma}$ gives, $\mu = \min\left\{\frac{\epsilon}{2\gamma\sigma^2}, \frac{1}{\gamma}\right\}$. Putting the step size in the first equation makes the objective $\Psi_M \leq \epsilon$ for any $M \geq \max\left\{\frac{4F\gamma}{\epsilon}, \frac{8F\gamma\sigma^2}{\epsilon^2}\right\}$. This implies $M = O(\frac{1}{\epsilon} + \frac{\sigma^2}{\epsilon^2})$ and completes the proof. $\qquad\square$

### B.1.1 Proof of Lemma 2

*Proof.* We can use the quadratric upper bound from the Lipschitz condition, the response in (2) and the successful gradient update step of (4) to obtain the following bound,

$$f(x_{k_{m+1}}) \leq f(x_{k_m}) - \mu\langle \nabla f(x_{k_m}), r_{k_m}\rangle + \frac{\mu^2\gamma}{2}\left(\|r_{k_m}\|^2\right)$$

$$f(x_{k_{m+1}}) \leq f(x_{k_m}) - \mu\langle \nabla f(x_{k_m}), \nabla f(x_{k_m}) + \eta_{k_m}\rangle + \frac{\mu^2\gamma}{2}\left(\|\nabla f(x_{k_m}) + \eta_{k_m}\|^2\right)$$

$$f(x_{k_{m+1}}) \leq f(x_{k_m}) - \left(\mu - \frac{\mu^2\gamma}{2}\right)\|\nabla f(x_{k_m})\|^2 - \left(\mu - \mu^2\gamma\right)\langle \nabla f(x_{k_m}), \eta_{k_m}\rangle + \frac{\mu^2\gamma}{2}\left(\|\eta_{k_m}\|^2\right).$$

Summing over the inequalities till $m = M$,

$$\left(\mu - \frac{\mu^2\gamma}{2}\right)\sum_{m=1}^{M}\|\nabla f(x_{k_m})\|^2 \leq f(x_{k_0}) - f(x_{k_{M+1}}) - \left(\mu - \mu^2\gamma\right)\sum_{m=1}^{M}\langle \nabla f(x_{k_m}), \eta_{k_m}\rangle + \frac{\mu^2\gamma}{2}\sum_{m=1}^{M}\|\eta_{k_m}\|^2$$

$$\leq f(x_{k_0}) - f^* - \left(\mu - \mu^2\gamma\right)\sum_{m=1}^{M}\langle \nabla f(x_{k_m}), \eta_{k_m}\rangle + \frac{\mu^2\gamma}{2}\sum_{m=1}^{M}\|\eta_{k_m}\|^2.$$

The last step is due to the function being lower bounded by $f^*$. This concludes the proof. $\qquad\square$

### B.1.2 Proof of Corollary 1

*Proof.* Taking expectation on (16) from Lemma 2 w.r.t. the noise terms $\eta_{k_0}, \eta_{k_1}, \ldots, \eta_{k_M}$,

$$\left(\mu - \frac{\mu^2\gamma}{2}\right)\sum_{m=1}^{M}\mathbb{E}\left[\|\nabla f(x_{k_m})\|^2\right] \leq \mathbb{E}\left[f(x_{k_0})\right] - f^* - \left(\mu - \mu^2\gamma\right)\sum_{m=1}^{M}\mathbb{E}\left[\langle \nabla f(x_{k_m}), \eta_{k_m}\rangle\right] + \frac{\mu^2\gamma}{2}\sum_{m=1}^{M}\mathbb{E}\left[\|\eta_{k_m}\|^2\right].$$

Now using assumption A2, $\mathbb{E}\left[\langle \nabla f(x_{k_m}), \eta_{k_m}\rangle | \eta_{k_0}, \ldots, \eta_{k_{m-1}}\right] = 0$ and by (A2) and definition of a successful gradient step, $\mathbb{E}\left[\|\eta_{k_m}\|^2\right] \leq \sigma_{k_m}^2 \leq \sigma^2$. The above equation therefore reduces to,

$$\left(\mu - \frac{\mu^2\gamma}{2}\right)\sum_{m=1}^{M}\mathbb{E}\left[\|\nabla f(x_{k_m})\|^2\right] \leq \mathbb{E}\left[f(x_{k_0})\right] - f^* + \frac{\mu^2\gamma M\sigma^2}{2}.$$

Denoting $F_0 = \mathbb{E}f(x_{k_m}) - f^*$, and using $\mu\gamma \leq 1$ we obtain,

$$\frac{\mu}{2}\left(2 - \mu\gamma\right)\sum_{m=1}^{M}\mathbb{E}\left[\|\nabla f(x_{k_m})\|^2\right] \leq F_0 + \frac{\mu^2\gamma M\sigma^2}{2} \implies \frac{\mu}{2}\sum_{m=1}^{M}\mathbb{E}\left[\|\nabla f(x_{k_m})\|^2\right] \leq F_0 + \frac{\mu^2\gamma M\sigma^2}{2}.$$

Multiplying by $2/\mu M$ on both sides we obtain the desired result,

$$\Psi_M \leq \frac{2F_0}{M\mu} + \mu\gamma\sigma^2.$$

$\qquad\square$

### B.2 Proof of Lemma 1

*Proof.* Let $\nu$ be a querying policy satisfying the constraint of (9), then

$$\Lambda \geq \limsup_{N\to\infty} \frac{1}{N}\mathbb{E}\left[\sum_{n=1}^{N} l(0, y_n^O = W_O)\mathbb{1}(a_n = 0, y_n^L > 0) \mid y_0\right].$$

If $\limsup_{N\to\infty} \frac{1}{N}\mathbb{E}\left[\sum_{n=1}^{N} \mathbb{1}(y_n^L = 0)|y_0\right] > 0$ then the queue is stable since the queue returns to the state $y_n^L = 0$ infinitely often and hence the Markov chain is also recurrent.

Otherwise if $\limsup_{N\to\infty} \frac{1}{N}\mathbb{E}\left[\sum_{n=1}^{N} \mathbb{1}(y_n^L = 0)|y_0\right] = 0$ the stability of the queue can be shown by proving that the average successful transmissions (denoted by $r_{y_0}(\nu)$) under the policy $\nu$ is greater than the average arrival rate $\delta M$,

$$r_{y_0}(\nu) = \liminf_{N\to\infty} \frac{1}{N}\mathbb{E}_\nu\left[\sum_{n=1}^{N} \Upsilon(u_n, y_n^O)\mathbb{1}(u_n \neq 0)|y_0\right] \geq \liminf_{N\to\infty} \frac{1}{N}\mathbb{E}_\nu\left[\sum_{n=1}^{N} \Upsilon_{\min}\mathbb{1}(u_n \neq 0)|y_0\right]$$

$$\geq \Upsilon_{\min}\left(1 - \limsup_{N\to\infty} \frac{1}{N}\mathbb{E}_\nu\left[\sum_{n=1}^{N} \mathbb{1}(u_n = 0, y_n^L > 0)|y_0\right]\right) \geq \Upsilon_{\min}\left(1 - \frac{\Lambda}{l(0, y^O = W))}\right) \geq \delta M.$$

Since the average successful learning rate is greater than the average query arrival rate, this induces a stable buffer, and due to Foster's Theorem, this induces a recurrent Markov chain. □

## B.3 Proof of Theorem 2

We state the following lemma which is key to prove Theorem 2.

**Lemma 3.** *(Monotonicy of value function $V$) The value function $V_n(y)$ is decreasing in number of queries left, $n$ and oracle state, $y^O$ and increasing in learner state, $y^L$.*

*Proof.* The strategy for proving the monotonicity of the value function with respect to the state space variables will be to use induction and assumptions about the cost function and the probability transition matrix.

The recursion for $V_{n+1}\left([y^O, y^L]\right)$ from (7) is,

$$V_{n+1}\left(y = [y^O, y^L]\right) = \min_{u\in\mathcal{U}} c(u, y^O) + \sum_{y^{O'}\in\mathcal{S}^O} \mathbb{P}(y^{O'}|y^O)\left(\Upsilon(y^O, u)V_n\left([y^{O'}, y^L - u]\right) + (1 - \Upsilon(y^O, u))V_n\left([y^{O'}, y^L]\right)\right),$$

where $V_0\left([y^{O'}, y^L]\right) = l(y^L)$.

**Monotonicity in $n$:** The first step of the induction, $V_1\left([y^O, y^L]\right) \leq V_0\left([y^{O'}, y^L]\right)$ can be shown as,

$$V_1\left([y^O, y^L]\right) = \min_{u\in\mathcal{U}} c(u, y^O) + \sum_{y^{O'}\in\mathcal{S}^O} \mathbb{P}(y^{O'}|y^O)\left(\Upsilon(y^O, u)l(y^L) + (1 - \Upsilon(y^O, u))l(y^L)\right)$$

$$\leq Q_1\left([y^O, y^L], 0\right) = l(y^L) = V_0\left([y^{O'}, y^L]\right).$$

Now let $V_n\left([y^{O'}, y^L]\right) \leq V_{n-1}\left([y^{O'}, y^L]\right)$, then using the recursion and the monotonicity of cost it is straightforward to show that it holds true for $n + 1$ since,

$$V_{n+1}\left([y^O, y^L]\right) \leq \min_{u\in\mathcal{U}} c(u, y^O) + \sum_{y^{O'}\in\mathcal{S}^O} \mathbb{P}(y^{O'}|y^O)\left(\Upsilon(y^O, u)V_{n-1}\left([y^{O'}, y^L - 0]\right) + (1 - \Upsilon(y^O, u))V_{n-1}\left([y^{O'}, y^L]\right)\right)$$

$$= V_n\left([y^O, y^L]\right).$$

**Monotonicity in $y^L$:** We use inductive reasoning again to prove that the value function is increasing in $y^L$. Note that $V_0\left([y^O, y^L]\right) = l(y^L)$ is increasing in $y^L$. Let $V_n\left([y^O, y^L]\right)$ be increasing in $y^L$. From the definition of $V_{n+1}\left([y^O, y^L]\right)$ it follows that $V_{n+1}$ is sum of increasing functions of $y^L$ hence it should also be increasing in $y^L$.

**Monotonicity in $y^O$:** Using the assumption of first order stochastic dominance on $\mathbb{P}(y^{O'}|y^O)$ we prove that $V_n\left([y^O, y^L]\right)$ is non-increasing in $y^O$. $V_0\left([y^O, y^L]\right) = l(y^L)$ is non-increasing in $y^O$. Assume $V_n\left([y^O, y^L]\right)$ is non-increasing.

Then, for $y^O > 1$, by monotonicity of $c$, $c(u, y^O) \leq c(u, y^O - 1)$. And by the first-order stochastic dominance assumption and induction assumption, $\sum_{y^{O'} \in \mathcal{S}^O} \mathbb{P}(y^{O'} | y^O) V_n \left( [y^{O'}, y^L] \right) \leq \sum_{y^{O'} \in \mathcal{S}^O} \mathbb{P}(y^{O'} | y^O - 1) V_n \left( [y^{O'}, y^L] \right)$.

Then for $y^O > 1$,

$$
V_{n+1} \left( y = [y^O, y^L] \right) = \min_{u \in \mathcal{U}} c(u, y^O) + \sum_{y^{O'} \in \mathcal{S}^O} \mathbb{P}(y^{O'} | y^O) \left( \Upsilon(y^O, u) V_n \left( [y^{O'}, y^L - u] \right) + (1 - \Upsilon(y^O, u)) V_n \left( [y^{O'}, y^L] \right) \right)
$$

$$
\leq \min_{u \in \mathcal{U}} c(u, y^O - 1) + \sum_{y^{O'} \in \mathcal{S}^O} \mathbb{P}(y^{O'} | y^O - 1) \left( \Upsilon(y^O, u) V_n \left( [y^{O'}, y^L - u] \right) + (1 - \Upsilon(y^O, u)) V_n \left( [y^{O'}, y^L] \right) \right)
$$

$$
= V_{n+1} \left( y = [y^O - 1, y^L] \right).
$$

$\square$

**Proof for the infinite horizon discounted MDP:** Since instantaneous cost $w$ is bounded. Hence the value function sequence,

$$
V_{n+1}^\beta \left( [y^O, y^L, y^E] \right) = \min_{u \in \mathcal{U}} \left[ w(u, y; \lambda) + \beta \sum_{\substack{y^{O'} \in \mathcal{S}^O \\ y^{E'} \in \mathcal{S}^E}} \mathbb{P}(y^{O'} | y^O) \mathbb{P}(y^{E'}) \left( \Upsilon(y^O, u) V_n^\beta \left( [y^{O'}, y^L + y^E - u, y^{E'}] \right) + \right. \right.
$$

$$
\left. \left. (1 - \Upsilon(y^O, u)) V_n^\beta \left( [y^{O'}, y^L + y^E, y^{E'}] \right) \right) \right],
$$

converges for any initial $V_0^\beta \left( [y^O, y^L, y^E] \right)$. Hence we choose a $V_0^\beta \left( [y^O, y^L, y^E] \right)$ which is increasing in $y^L, y^E$ and decreasing in $y^O$. Note that by assumptions on $c$ and $l$, $w(u, y; \lambda)$ is decreasing in $y^O$ and nondecreasing in $y^L$. Therefore by induction $V_n^\beta \left( [y^O, y^L, y^E] \right)$ is increasing in $y^L$ and $y^E$. And by assumption of first order stochastic dominance on $\mathbb{P}(y^{O'} | y^O)$ it is easy to see that $V_n^\beta \left( [y^O, y^L, y^E] \right)$ is decreasing in $y^O$. Therefore $V^\beta \left( [y^O, y^L, y^E] \right) = V_\infty^\beta \left( [y^O, y^L, y^E] \right)$ is increasing in $y^L$ and $y^E$ and decreasing in $y^O$.

We use this result on $V^\beta \left( [y^O, y^L, y^E] \right)$ in the discussion of structural results on the infinite horizon average cost MDP.

We now prove Theorem 2:

*Proof.* To show that the optimal discounted cost policy is monotonically increasing in learner state $y^L$, we will prove inductively that $Q_N \left( [y^O, y^L], u \right)$ is submodular in $(y^L, u)$ for all $y^L \geq 1$. In other words, we prove that,

$$
Q_{n+1} \left( [y^O, y^L], 1 \right) - Q_{n+1} \left( [y^O, y^L], 0 \right) = c(1, y^O) - c(0, y^O) + \sum_{y^{O'} \in \mathcal{S}^O} \mathbb{P}(y^{O'} | y^O) \Upsilon(y^O, 1)
$$

$$
\times \left[ V_n \left( [y^{O'}, y^L - 1] \right) - V_n \left( [y^{O'}, y^L] \right) \right],
$$

is monotonically decreasing in the learner state $y^L$ for $y \geq 1$ for all $n \geq 0$ for a suitable initialization. This is a sufficient condition for a monotone threshold policy since if the state action ($Q$) decreases monotonically. It will change its sign over the learner state space $\mathcal{S}^L$ only once, and the action 0 will be optimal until a certain value of $y^L$ and the action 1 will be optimal otherwise.

For illustration, in a finite horizon case, we assume the oracle state is fixed, and with abuse of notation, write the Q function only in terms of the learner state . Supermodularity of the Q function in $(y^L, u)$ is satisfied when . Supermodularity implies increasing differences in with increasing . The optimal action with queries remaining can be written as . Therefore, if at some , the optimal action is 0, then for , the optimal action has to be since the difference between the Q function only decreases. Hence, inductively, in can be shown that for , the optimal action will be 0. Similarly, it can be shown that if at some , the optimal action is 1, then for all , the optimal action would have to be 1. This gives rise to the threshold structure of the policy. This was an illustrative explanation of the concept of supermodularity, which has been used to show threshold policy results in previous work Ngo & Krishnamurthy (2010)

and can be extended to the discounted Lagrangian cost MDP case. Applying a policy trained by the policy gradient method to a stochastic gradient method can be difficult, due to the changing environment.

$Q_{n+1}\left([y^O, y^L], 1\right) - Q_{n+1}\left([y^O, y^L], 0\right)$ has increasing differences in $y^L$ if, $V_n\left([y^O, y^L]\right)$ has increasing difference in $y^L$. We prove this inductively. By assumption of integer convexity, $V_0\left([y^O, y^L]\right) = l(y^L)$, has increasing differences in $y^L$. Assume $V_n\left([y^O, y^L]\right)$ has increasing differences in $y^L$ then $Q_{n+1}\left([y^O, y^L], u\right)$ is submodular in $(y^L, u)$. We will now show that $V_{n+1}\left([y^O, y^L]\right)$ has increasing differences in $y^L$, i.e.

$$V_{n+1}\left([y^O, y^L + 1]\right) - V_{n+1}\left([y^O, y^L]\right) - \left(V_{n+1}\left([y^O, y^L]\right) - V_{n+1}\left([y^O, y^L - 1]\right)\right) \geq 0. \tag{18}$$

Now let $Q_{n+1}\left([y^O, y^L], u_1\right) = V_{n+1}\left([y^O, y^L + 1]\right)$, $Q_{n+1}\left([y^O, y^L], u_2\right) = V_{n+1}\left([y^O, y^L]\right)$ and $Q_{n+1}\left([y^O, y^L], u_3\right) = V_{n+1}\left([y^O, y^L - 1]\right)$ for some actions $u_1, u_2$ and $u_3$. Now (18) can be written as,

$$Q_{n+1}\left([y^O, y^L + 1], u_1\right) - Q_{n+1}\left([y^O, y^L], u_1\right) - \left(Q_{n+1}\left([y^O, y^L], u_1\right) - Q_{n+1}\left([y^O, y^L - 1], u_2\right)\right) \geq 0 \iff$$

$$\underbrace{\left(Q_{n+1}\left([y^O, y^L + 1], u_1\right) - Q_{n+1}\left([y^O, y^L], u_1\right)\right)}_{A} - \underbrace{\left(Q_{n+1}\left([y^O, y^L], u_1\right) - Q_{n+1}\left([y^O, y^L], u_2\right)\right)}_{\text{By optimality} \geq 0}$$

$$- \underbrace{\left(\left(Q_{n+1}\left([y^O, y^L], u_2\right) - Q_{n+1}\left([y^O, y^L], u_3\right)\right)\right)}_{\text{By optimality} \geq 0} - \underbrace{\left(\left(Q_{n+1}\left([y^O, y^L], u_3\right) - Q_{n+1}\left([y^O, y^L - 1], u_3\right)\right)\right)}_{B} \geq 0.$$

Now rearranging the terms for $A$,

$$\begin{aligned} A = \sum_{y^{O'} \in \mathcal{S}^O} \mathbb{P}(y^{O'}|y^O) \times \Big[ \Upsilon(y^O, u_2)] \left(V_n\left([y^{O'}, y^L]\right) - V_n\left([y^{O'}, y^L - 1]\right)\right) \\ + (1 - \Upsilon(y^O, u_2)) \left(V_n\left([y^{O'}, y^L + 1]\right) - V_n\left([y^{O'}, y^L]\right)\right) \Big] \geq \\ \sum_{y^{O'} \in \mathcal{S}^O} \mathbb{P}(y^{O'}|y^O) \times \left[ V_n\left([y^{O'}, y^L]\right) - V_n\left([y^{O'}, y^L - 1]\right)\right] \geq B. \end{aligned}$$

The second last inequality is due to induction on $V_n\left([y^O, y^L]\right)$ and the last inequality follows from similar expansion on $B$ and induction hypothesis. This theorem can be straightforwardly extended to infinite horizon discounted MDP. $\square$

## B.4 On threshold structure of average Lagrangian cost optimal policy

To show that the optimal policy of the unconstrained average cost MDP has a threshold structure, we first state the following lemma (Sennott, 1989).

**Lemma 4.** *Let $(\beta_k)$ be any increasing sequence of discount factors, s.t., $\lim_{k \to \infty} \beta_k = 1$. Let $(\nu^*_{\beta_k})$ be the associated sequence of discounted optimal stationary policies. There exist a subsequence $(\alpha_k)$ of $(\beta_k)$ and a stationary policy $\nu$ that is the limit of $(\nu^*_{\alpha_k})$.*

An optimal policy given by Lemma 4 is an average cost optimal policy under suitable assumptions (Sennott, 1989). We verify these assumptions and characterize the average cost optimal policy in the following theorem:

**Theorem 4.** *Any stationary deterministic policy $\nu$ given by Lemma 4 is an average cost optimal policy. In particular there exists a constant $\psi = \lim_{\beta \to 1}(1 - \beta)V^\beta(y)$ for every $y$, and function $\Psi(y)$ with $-N \leq \Psi(y) \leq M_y$, such that,*

$$\psi + \Psi(y) = \min_{u \in \mathcal{U}} \left\{ w(u, y; \lambda) + \sum_{y' \in \mathcal{S}} \mathbb{P}(y'|y, u)\Psi(y) \right\}.$$

*Furthermore, the stationary policy is average cost optimal with an average cost $\psi$.*

*Proof.* For any stationary policy $\nu$ to be average cost optimal, the following assumptions need to be satisfied (Sennott, 1989):

- Assumption 1: For every state $y$ and discount factor $\beta$, the optimal discounted cost $V^\beta(y)$ is finite.

- Assumption 2: There exists $N \geq 0$ such that, $-N \leq \Psi^\beta(y) \stackrel{\Delta}{=} V^\beta(y) - V^\beta(0)$ where 0 is a reference state.

- Assumption 3: There exists $M_y \geq 0$, such that, $\Psi^\beta(y) \leq M_y$ for every $y$ and $\beta$. For every $y$ there exists, $u$ such that $\sum_{y' \in \mathcal{S}} \mathbb{P}(y'|y, u) < \infty$.

- Assumption 3': Assumption 3 holds and $\sum_{y' \in \mathcal{S}} \mathbb{P}(y'|y, u) M_y < \infty$.

For a reference state $0 = [0, W, 0]$, the policy of always transmitting induces a stable buffer, and the expected time and cost for the first passage to state $0$ are finite. Therefore by Proposition 5i) and 4ii) of Sennott (1989) and from Ross (2014) Assumption 1 and 3 are satisfied. Assumption 3' is satisfied by the probability transition given in (8). Assumption 2 is satisfied because $V^\beta$ is increasing in $y^L$, $y^E$ and decreasing in $y^O$ and therefore $V^\beta(y) \geq V^\beta(0) \ \forall y \in \mathcal{S}$ as shown in Lemma 3. $\qquad \square$

Due to the above lemma and theorem and using the discussion in the proof of Lemma 3, the average cost optimal policy inherits the monotone threshold structure of the discounted optimal policy.

