# OpenReview forum: "Controlling Federated Learning for Covertness"
_TMLR — Accepted by TMLR_

### Review · Reviewer_TFyX · 2023-08-23

**Summary Of Contributions:**

In this paper, a learner seeks to minimize a function f via gradient descent by querying a distributed oracle that provides noisy gradient evaluation. However, the learner seeks to hide the argmin of f from a malicious eavesdropper.

Let me mention that I am not an expert in federated learning but I do know a bit about covertness in the context of information theory.

**Audience:**

Yes

**Broader Impact Concerns:**

NIL

**Claims And Evidence:**

Yes

**Requested Changes:**

The reviewer, however, has the following major concerns about the submission.

1) There does not seem to be a theoretical guarantee for the Structured Policy Gradient algorithm in Section 3.4.

2) When researchers in information theory use the word "covert", it usually means something different. Please search "covert communications" and look at this paper.
https://ieeexplore.ieee.org/document/7407378/
Herein a major consideration is to quantify the **tradeoff** between the level of covertness and throughput. In the authors' investigations, they should also quantify the tradeoff between the convergence rate and the level of covertness (e.g., how well the argmin f is actually hidden). This is not done in the present submission.

**Strengths And Weaknesses:**

The formulation and questions asked in this paper seem to be interesting and meaningful. The experimental evaluation is useful. The derivation of a Markov decision process as a means to control the stochastic gradient descent algorithm is also fairly novel.

---

> ### Author Response · Authors · 2023-08-25
> **Reply to reviewer TFyX**
>
> Thank you for taking the time to read and understand our paper and providing us with useful feedback on it. We have tried our best to address your concerns:
>
> 1. It is difficult to obtain theoretical guarantees on a stochastic policy search algorithm, however due to our structural results the search space reduces drastically ( 6|Y_O| +1 compared to 6L|Y_O| where L is the length of the finite queue approximation ) which makes our algorithm computationally efficient. We have addressed this point in Appendix A.6 along with supplementary results using a constant step size in SPGA algorithm.
> 2. For the finite horizon case we have added the effect of the percent of queries obfuscated on the convergence (inversely proportional to log dependence) and covertness for the proportional sampling estimator considered in the paper (step function dependence) in Appendix A.7. We provide a discussion on  the current theoretical results on the tradeoff but these results are for a convex function. For a general non convex function, the bounds on the minimum number of queries for a given probability of covertness are difficult to obtain in general.
> 3. We understand that the term covert might cause confusion with the existing literature in Information Theory and are open to changing it in the final submission to avoid confusion with the other reviewers. We think the original "learner-private" from [Tsitsiklis, 2021].
>
> We appreciate your feedback and would appreciate further comments and discussion on these points, and the paper in general.

---

> > ### Comment · Reviewer_TFyX · 2023-08-25
> >
> > Thank you for your replies. They sound sensible, though it would have been good to have some theoretical results that elucidate the tradeoff between covertness and convergence rates. I'm not an expert in federated learning, and my review is nothing more than a "best guess".

---

> > > ### Comment · Action_Editors · 2023-08-25
> > > **RE: Federated Learning**
> > >
> > > >  I'm not an expert in federated learning, and my review is nothing more than a "best guess".
> > >
> > > Please don't worry about not knowing much about FL, this is completely fine. One of the reviewers specializes in FL, so I hope to have feedback on the FL side from them. If you can give a good evaluation of other aspects of the work, that would be great.

---

### Review · Reviewer_N6BD · 2023-09-09

**Summary Of Contributions:**

The paper studies the covert optimization problem in federated learning in which a learner dynamically decides to query a stochastic gradient oracle. The authors model the decision-making problem of choosing stochastic gradients as a (constrained) Markov decision process and present several structural properties of the underlying optimal policies. To find an optimal policy, the authors propose a policy gradient method and demonstrate the performance of proposed methods in experiments.

**Audience:**

Yes

**Claims And Evidence:**

Yes

**Requested Changes:**

Here are some other questions:

- What is the definition of $P$ in Equation (1)?

- It seems difficult to determine $M$ in MDPs. How this can be handled in policy gradient algorithms and experiments.

- It is useful to illustrate the super modular structure more and its implications for the optimal policy?

- Applying a policy trained by the policy gradient method to a stochastic gradient method can be difficult, due to the changing environment. How does the policy trained by the policy gradient method generalize?

**Strengths And Weaknesses:**

Strengths:

- The authors apply the classical Markov decision process (MDP) to model the query of stochastic gradient oracle. This provides a general framework to study the eavesdropper issue in federated learning.

- The authors show that the optimal policies of proposed MDPs have a threshold structure that can be used to reduce the policy search space. By approximating the policy using sigmoid functions, the authors propose a structured policy gradient to learn an optimal query policy. This is a useful application of policy gradient methods in federated learning.

- The authors also provide experiments to show the performance of the proposed methods. The hate speech classification seems to be a new application of covert optimization.

Weaknesses:

- Modeling the stochastic gradient with controlled oracles as a MDP problem is intuitive. Can the authors provide more rigorous statements? Any simple examples of stochastic gradient methods?

- Assumptions made about the proposed MDPs can impose the limitations. It is important to illustrate why the stochastic gradient method is the case.

- The non-asymptotic convergence of stochastic gradient method is quite standard in theory. However, non-asymptotic convergence is absent for the proposed policy gradient method. It is also less discussed the computational complexities of the algorithm.

- Performance of the proposed policy is similar as greedy policy. Pros and cons of the propopsed method needs to be clarified.

---

### Review · Reviewer_Ai6d · 2023-11-21

**Summary Of Contributions:**

This paper considers covert FL, where the learner have two goals simultaneously: (1) learn the minimizer of a function $f$ based on the noisy gradients from distributed oracles by sending a query $q$, and (2) hiding the minimizer of $f$ from a malicious eavesdropper.

The authors considered this problem as controlling the SGD by a policy gradient algorithm that exploits the policy structure. The authors showed that the optimal policy has a monotone threshold structure. This paper also proposes an efficient policy gradient algorithm, which well balances between accuracy and robustness (against eavesdropper) in practical scenarios including the hate speech classification task.

**Audience:**

Yes

**Claims And Evidence:**

Yes

**Requested Changes:**

Probably it is better to improve the readability of the paper. Making a table summarizing important notations would be helpful. In the caption of figure/table, it is better to what each mathematical notations is (e.g., \phi, \theta in Fig.4)

**Strengths And Weaknesses:**

Strength
- The authors provided a proper motivation for such problem in the paper. Especially, the description on the problematic scenario when the eavesdropper behave maliciously based on the revealed information on the hate speech classification model
- This paper provides some theoretical and empirical results on the optimal policy

---

> ### Author Response · Authors · 2023-11-22
> **Response to Reviewer Ai6d**
>
> We greatly appreciate your time and effort in providing insightful and encouraging comments that have helped us improve our work. In response to your suggestions, we have incorporated a summary of the notation used in our paper in the latest revision. This improves the readability of the text and provide a quick reference for readers to understand the symbols and abbreviations used throughout the paper.  We look forward to any additional feedback you may have. Thank you once again for your review of our work.

---

### Comment · Action_Editors · 2024-01-10
**Some minor issues**

Dear authors,

While going through the paper, I found a few small issues that I'd like you to fix, they are listed below. There is also something that concerns me about the assumptions. There is the following statement in the paper:
> Assumptions (A1) and (A2) are standard regularity assumptions when analyzing query complexity results of stochastic
gradient algorithms (Ghadimi & Lan, 2013).

This statement seems misleading to me. (Ghadimi & Lan, 2013) only assume smoothness and bounded variance of the stochastic gradients, whereas this submission assumes a compact domain in (A1) and bounded stochastic gradients in (A2). I ask the authors to update this passage and be honest about the restrictive nature of the assumptions.

### Minor issues
* (A1) writes $|\nabla f(z) - \nabla f(x)|$ to denote the norm of the grad differences, whereas the text uses $\Vert \nabla f(x)\Vert$, e.g., in equation (3). Please make the notation consistent, preferably using $\Vert \cdot \Vert$.
* The use of footnotes is excessive, there are 24 footnotes in the main part of the paper, and four of them on page 4. Some footnotes, for example, footnotes 5 and 6, would be better written in the text, so I suggest the authors check again if the footnotes are necessary.
* Please check the punctuation in the equations. For example, equations (11), (15), (16), the last equation on page 25 should all end with "."
* "first order stochastic dominance" -> "first-order stochastic dominance"
* Appendix A.6 "atleast" -> "at least"

---

> ### Author Response · Authors · 2024-01-11
> **Response to the minor issues**
>
> Dear Action Editor,
>
> We value your detailed feedback on the assumptions and thank you for pointing out the minor issues. We have implemented the following changes to address them:
>
> **Assumption:**
>
> We acknowledge your concern regarding the statement about Assumption (A1) and (A2) and we agree that it was misleading. In our initial presentation of Assumptions (A1) and (A2), we aimed to acknowledge their familiarity within the field. However, we readily admit that this phrasing downplayed the limitations inherent to these specific assumptions, especially the restrictive nature of the compactness and boundedness conditions. We have revised the passage to be more transparent about the restrictive nature of our assumptions and how the assumptions can be weakened. We have replaced the statement with:
>
> > Assumptions (A1) and (A2) are regularity assumptions for analyzing query complexity results of the stochastic gradient descent. There are limitations to these assumptions when compared to the standard first-order gradient descent analysis (Ghadimi & Lan, 2013): Assumption (A1) assumes compactness of the domain and Assumption (A2) implicitly assumes boundedness of the gradients. The first limitation is a matter of convenience and the domain can also be chosen as $\mathbb{R}^d$ and the second limitation provides ease in manipulating the descent lemma to prove the finite sample bound. Assumption (A2) can be weakened with a bound on just the noise variance $\mathbb{E}\left[||\eta_k||^2\right] \leq \sigma_k^2$ to obtain the standard finite sample bound of $O(1/\epsilon + \sigma^2/\epsilon^2)$ (Theorem 4 in Ajalloeian & Stich (2021)).
>
> We would greatly appreciate any additional feedback you may have on the revised passage regarding the assumptions. We believe the other aspects align well with the cited reference, but we remain open to further improving the accuracy of the text. Your continued insights are instrumental in strengthening our work.
>
> **Minor Issues:**
>
> - We have adopted $||\cdot||$ notation consistently throughout the paper, including replacing $|∇f(x) - ∇f(y)|$ with $||∇f(x) - ∇f(y)||$.
> - We have carefully reviewed the footnotes and incorporated several, like footnotes 5 and 6, into the main text. We have retained only 10 (out of the original 24) essential footnotes that provide relevant clarifications without disrupting the flow of the text.
> - We have corrected the punctuation in all the equations including (11), (15), (16), and the last equation on page 25, ensuring they all end with periods.
> - We have corrected the typo "first order stochastic dominance" to "first-order stochastic dominance" in the appropriate places.
> - We have corrected the typo "atleast" to "at least" in Appendix A.6.
>
> We are grateful your assistance in making our work stronger and the revisions improve the clarity of our paper.
> Again, thank you for your insightful feedback on the assumptions. Your point is well-taken, and we appreciate you calling this to our attention.
>
> Warm regards,
>
> Authors

---

> > ### Comment · Action_Editors · 2024-01-16
> >
> > Thanks for fixing the small issues. I have submitted a positive recommendation for your paper asking for a small revision to encourage you to go through the paper again to proofread it and fix any remaining issues, such as typos. In case you can incorporate the mentioned changes to weaken the assumption and make it just about the gradient variance, that will be an improvement too, but it is completely optional.

---

> > > ### Author Response · Authors · 2024-01-16
> > > **Thanks for the recommendation**
> > >
> > > Dear Action Editor
> > >
> > > We sincerely appreciate your time and effort in reviewing our paper. Thank you for your positive recommendation and  your help in improving the quality of our research. We will proofread the paper thoroughly to address any remaining typos or inconsistencies and would try our best to ensure that the paper meets the standards of TMLR. Additionally, we will incorporate your feedback on the assumption. We also plan to add a subsection in the appendix to an alternative convergence rate result without the assumption on the gradient norm.
> > >
> > > We will make the necessary changes and submit a revised camera ready version within a couple of days. We also wanted to inquire about the video submission, can it be added post submitting the camera-ready version, and what is a suitable length for the video?
> > >
> > > Thank you again for your insightful feedback and your positive recommendation.
> > >
> > > Sincerely,
> > >
> > > Authors

---

> > > > ### Comment · Action_Editors · 2024-01-17
> > > >
> > > > Regarding video, I’m not sure how this works. It should be mentioned in the camera ready version instructions, but if not, you can ask editors-in-chief directly using their email.

---

> > > > > ### Author Response · Authors · 2024-01-31
> > > > > **Camera Ready Version Submitted**
> > > > >
> > > > > Dear Action Editor,
> > > > >
> > > > > Thank you once again for accepting our manuscript for publication in TMLR. We appreciate the insightful comments and suggestions, which have significantly improved the quality of our work. We are pleased to submit the revised camera-ready version of manuscript incorporating the feedback.
> > > > >
> > > > > In this revised version, we have made the following key changes:
> > > > >
> > > > > - **Domain Change and Alternative Approach:** We have expanded the analysis to the domain of $\mathbb{R}^d$ and provided an alternative approach with a less restrictive assumption on gradients norm in Appendix B.1 that does not require bounded gradients. This addition broadens the applicability of our work and offers readers a complementary perspective.
> > > > > - **Enhanced Text Clarity:** We have reorganized and rewritten the text, like contributions 1 and 2 to improve readability and flow.
> > > > > - **Notational Consistency:** We hava made the notation more consistent (different notation for L and G) throughout the manuscript for consistency and clarity. We have also addressed small notational inconsistencies in the proof, ensuring the mathematical arguments are rigorous and accurate.
> > > > > - **Grammar:** We have implemented grammar changes and improved sentence structure for better comprehension.
> > > > > - **Terminology Change:** We have replaced the phrase "optimal policy solving the CMDP" with "optimal solving for the CMDP" for conciseness.
> > > > >
> > > > > We believe these revisions, particularly the expanded domain and alternative approach, strengthen the manuscript. We have also addressed the reviewers' specific points in detail, as summarized in the  response letter in our previous comment ([link](https://openreview.net/forum?id=g01OVahtN9&noteId=hssC0QoF5T)).
> > > > >
> > > > > Let us know if there are other changes in the manuscript or formalities from our side and we'd be happy to do the needful. Thank you again for your time and consideration.
> > > > >
> > > > > Sincerely,
> > > > >
> > > > > Authors

---

> > > > > > ### Comment · Action_Editors · 2024-02-07
> > > > > > **There are issues with the updated proof**
> > > > > >
> > > > > > Dear authors,
> > > > > > I looked at the proof in Appendix B.1 and I see some issues.
> > > > > > 1. The indices in the second line of (16) are messed up, sometimes you write $k$ and sometimes $k_m$, this should be fixed.
> > > > > > 2. If I'm not mistaken, in the third line of equation (16), the sign before $\mu_{k_m} \langle \nabla f(\hat x_{k_m}), \eta_{k_m} \rangle$ should be $-$, not $+$.
> > > > > > 3. In the third line of (16), you have replaced $\Vert\nabla f(\hat x_{k_m}) + \eta_k\Vert^2$ with $\sigma^2$. However, you can't upper bound the variance without taking expectation first, as your assumption is on the expectation of the squared norm.
> > > > > > 4. In the lines to follow, you only have the expectation next to the gradient norm terms, but not next to the functional values, which is wrong.
> > > > > > 5. In the first line of (16) and what follows, you use $L$-smoothness, whereas the rest of the paper seems to assume that the objective is $\gamma$-smooth (different notation).
> > > > > >
> > > > > > To add to this:
> > > > > > 1. Why do you even need to maintain two proofs? The one with the relaxed assumptions seems to be strictly better.
> > > > > > 2. I asked you to make the notation for norms consistent, however, in Appendix B.2, Assumption (A1), you still write $|z-x|$ instead of $\Vert z - x\Vert$.
> > > > > >
> > > > > > These issues all seem to be easy to fix, so please update the paper.

---

> > > > > > > ### Author Response · Authors · 2024-02-08
> > > > > > > **Official comment by the Authors**
> > > > > > >
> > > > > > > Dear Action Editor,
> > > > > > >
> > > > > > > We are grateful for your thorough review and insights you've provided regarding the proofs in Appendix B. We apologize for the inconsistency in the notation that resulted from the changes we made (we have now ensured that the notation is consistent and all the lines of the proofs make logical sense).
> > > > > > >
> > > > > > > - We had intially kept both the proofs since we didn’t change the main text (Theorem 1) and wanted to illustrate how the alternative result could be proven. However we have now updated Theorem 1 and assumptions A1-2 in the main text, to reflect the relaxed assumptions and the corresponding finite-sample result.
> > > > > > > - As a consequence we have removed the previous proof and have kept the updated proof. We have also made small changes to the proof to ensure better readability and understanding.
> > > > > > >
> > > > > > > We are committed to maintaining the clarity and accuracy in our work and regret any inconvencience caused due to our oversight. We appreciate your understanding and patience and are grateful for the opportunity to improve our manuscript. Please let us know if there are any further changes and we’d be happy to incorporate them.
> > > > > > >
> > > > > > > Best regards,
> > > > > > >
> > > > > > > Authors

---

### Decision · Action_Editor_LFLU · 2024-01-10

**Recommendation:** Accept with minor revision

**Comment:**

## Summary and strengths
All reviewers were positive about the paper and pointed out that the problem is well-motivated, reasonably studied from the theoretical point of view, and given a sufficient experimental evaluation.

## Weaknesses
In short, the theory is somewhat limited, and only one problem is studied numerically.

**Extra details**. One weakness pointed out by Reviewer N6BD is that the paper's guarantees are asymptotic. The authors explained the difficulty of getting non-asymptotic guarantees in their response and Appendix A.6. While I understand the technical difficulty, it is, nevertheless, of higher interest to the TMLR community to have non-asymptotic guarantees.

Reviewer N6BD also pointed out that the assumptions are a bit limiting, which I also agree with. The assumptions made about the optimization aspects (compact domain, bounded stochastic gradients) are not what the modern theory of SGD uses.

**Audience:**

The paper considers an interesting and practical formulation of Federated Learning (FL), which would be of interest to those studying privacy and optimization aspects of FL.

**Claims And Evidence:**

The paper does exactly what it says: the authors properly formalize the objective, propose an algorithm, and demonstrate its performance in a practical application.